# Access to water and sanitation among people with disabilities: results from cross-sectional surveys in Bangladesh, Cameroon, India and Malawi

Islay Mactaggart,[1] Wolf-Peter Schmidt,[2] Kristof Bostoen,[3] Joseph Chunga,[4] Lisa Danquah,[5] Amal Krishna Halder,[6] Saira Parveen Jolly,[7] Sarah Polack,[1] Mahfuzar Rahman,[7] Marielle Snel,[8] Hannah Kuper,[1] Adam Biran[2]

For numbered affiliations see end of article.

**Correspondence to**
Dr Islay Mactaggart;
islay.mactaggart@lshtm.ac.uk

## ABSTRACT

**Objectives** To assess access to adequate water, sanitation and hygiene (WASH) among people with disabilities at the household and individual level.

**Design** Cross-sectional surveys.

**Setting** Data were included from five district-level or regional-level surveys: two in Bangladesh (Bangladesh-1, Bangladesh-2), and one each in Cameroon, Malawi and India.

**Participants** 99 252 participants were sampled across the datasets (range: 3567–75 767), including 2494 with disabilities (93–1374).

**Outcome** Prevalence of access to WASH at household and individual level.

**Data analysis** Age/sex disaggregated disability prevalence estimates were calculated accounting for survey design. The Unicef/WHO Joint Monitoring Programme definitions were used to classify facilities as improved/unimproved. Multivariable logistic regression was undertaken to compare between households with/without a person with a disability, and to identify predictors of access among people with disabilities.

**Results** There were no differences in access to improved sanitation or water sources between households with/without members with disabilities across the datasets. In Bangladesh-2, households including a person with a disability were more likely to share facilities with other households (OR 1.3, 95% CI 1.1 to 1.5). Households with people with disabilities were more likely to spend >30 min (round-trip) collecting drinking water than households without in both Cameroon (OR 1.8, 95% CI 1.0 to 3.4) and India (OR 2.3, 95% CI 1.2 to 4.7). Within households, people with disabilities reported difficulties collecting water themselves (23%–80% unable to) and accessing the same sanitation facilities as other household members, particularly without coming into contact with faeces (up to 47% in Bangladesh-2). These difficulties were most marked for people with more severe impairments.

**Conclusions** People with disabilities may not have poorer access to WASH at the household level, but may have poorer quality of access within their households. Further programmatic work is needed to ensure WASH facilities are inclusive of people with disabilities.

## Strengths and limitations of this study

► The surveys were relatively large and conducted in geographically different areas.
► Detailed information was collected on access to water, sanitation and hygiene (WASH) and the presence of disabilities in different settings.
► Different methods and disability definitions were used across the surveys, making comparison of prevalence difficult.
► Each survey was conducted at the state or regional level and the estimates derived therefore do not constitute national estimates.
► Comparative data on indicators of the quality of access to WASH among people without disabilities were not collected in any of the studies (eg, could hygiene facilities be used without incurring pain).

## INTRODUCTION

Access to clean water, adequate sanitation and hygiene (WASH) for the prevention or mitigation of faeco-oral infections is a fundamental component of public health.[1–4] This is underlined by the focus of Sustainable Development Goal 6: ensure access to water and sanitation for all.[5] 2015 estimates suggest that 91% of the global population now use an improved drinking water source (protected from outside contamination) and 68% use an improved sanitation facility (one that separates human excreta from human contact), with 1.5% of the current global disease burden attributed to unimproved water and sanitation.[2 6–8] However, these aggregate global figures may mask vulnerabilities within specific populations.[9 10]

Persons with disabilities are described in the United Nations Convention on the Rights of Persons with Disabilities as those who 'experience long-term physical, mental, intellectual or sensory impairments which,

in interaction with various barriers, may hinder their full and effective participation in society on an equal basis with others'.[11] Persons with disabilities constitute up to 15% of the global population, and are asserted to experience poorer access to WASH, although few quantitative data exist.[12] These assertions are grounded in the limited, but growing, body of evidence that associates disability status with greater risk of widespread exclusion from education and economic productivity, alongside poor access to healthcare in multiple countries and contexts.[12–14] A qualitative review by Groce *et al* lists potential barriers to WASH among persons with disabilities in the spheres of technical access barriers (such as facility structure and distance to facilities), social barriers related to stigma or abuse; and communication barriers[15,16]. The presence of social barriers, alongside the inter-relationships between WASH and poverty, and poverty and disability, may also impact on WASH access within the household. Several censuses and household surveys have collected data on both WASH access and disability; however, few surveys have reported on the association. Moreover, WASH access is usually only measured at the household level, and many previous studies have used inconsistent definitions of disability. Comparable quantitative data therefore remain lacking on whether persons with disabilities live in households that have poorer access to WASH, and whether persons with disabilities have poorer access to WASH than other members of their household.

To address these gaps in knowledge, we analysed data from five cross-sectional, population-based studies conducted at the district or regional level that had collected data on disability and access to, and experience of, WASH. These studies were conducted in: Cameroon, Malawi, India and Bangladesh (two discrete studies). This secondary analysis was opportunistic, based on data available. However, the inclusion of data from two African and two South-East Asian settings allowed a focus on the regions with lowest access to improved facilities globally.[17]

We explored 1) whether households including persons with disabilities have different access to WASH compared with households without disabled members; 2) whether persons with disabilities have different access to WASH compared with other members of their household and 3) which factors predict access to WASH among persons with disabilities.

## METHODS

We analysed data from five cross-sectional, population-based studies conducted at the district or regional level that had collected data on disability and access to WASH. These studies were conducted between 2013 and 2015 in Cameroon, Malawi, India and Bangladesh (two discrete studies).

Table 1 provides a comparative overview of the datasets in each study, including definitions of disability and WASH used.

Each of the studies comprised a single-district or multiple-district, population-based survey. One of the studies conducted a census, that is, included all individuals living within a defined geographical area (Bangladesh-2). The remaining four studies used cluster sampling to select a random, representative sample of participants from a defined population. The samples were not nationally representative and are referred to by their country of origin for brevity. In all settings, a household-level questionnaire was administered and additional questionnaire modules were used for people identified through the survey as having disabilities. In Cameroon, India and Malawi, a nested case-control study was undertaken, whereby cases with disabilities and age-sex-cluster-matched controls without disabilities were selected for further structured interview. In all settings, written informed consent was collected before the questionnaire was administered, and ethical approval was obtained from the relevant boards. Ethical committees that approved each study are provided in online supplementary file 1.[18]

### Disability assessment

Four of the five surveys (Bangladesh-2, Cameroon, India and Malawi) used iterations of the Washington Group Short Set (WGSS) questions to screen for disability in the population (see online supplementary appendix 1 for full question sets).[19 20] This questionnaire assesses disability status via reported limitation in core functional domains (such as seeing, hearing and self-care) on a 4-point scale ('no difficulty', 'some difficulty', 'a lot of difficulty' and 'cannot do at all'). Participants reporting 'a lot of difficulty' or 'cannot do at all' for any item in the questionnaire were considered to have a disability in each of the four studies. In Cameroon and India, the Washington Group screen was supplemented with clinical screening to detect the presence of moderate or severe physical, visual or hearing impairments or severe health conditions (epilepsy and depression) using prevalidated screening methods and thresholds.[21–24] Conversely, Bangladesh-1 used a binary screen for disability, whereby disability status was reported by the household head for each member of their household as 'disability—yes or no'. If the household head reported that a member of their household had a disability, they were asked to identify the type of disability (eg, vision, hearing). Type of disability in Bangladesh-1 also included 'age-related', defined in the study as associated with the process of ageing.

### WASH assessment

The WASH question sets used in each study were adaptations of the WHO/United Nations Children's Fund Joint Monitoring Programme (JMP) *Core Questions on Drinking Water and Sanitation for Household Surveys,* and the JMP draft intrahousehold question set (see online supplementary appendix 1 for full question sets).[25 26] Information about WASH at the household level was collected through interview with either the head of household (Bangladesh-1,

**Table 1**  Study methods

| | Bangladesh-1 (icddr,b) | Bangladesh-2 (IRC-BRAC) | Cameroon (ICED, LSHTM) | India (ICED, LSHTM) | Malawi (collaboration*) |
|---|---|---|---|---|---|
| Geographical location | 177 subdistricts (upazillas) across Bangladesh (estimated population size 750 000) | Rangpur and Rajshahi Districts, Bangladesh (estimated population size 34 149 858) | Fundong Health District, North-West Region (estimated population size 125 604) | Mahbubnager District, Telegana State (estimated population size 4 053 028) | Rumphi District, Northern Region (estimated population size 203 054) |
| Year of data collection | 2014 | 2015 | 2013 | 2014 | 2014 |
| Sample size (response rate %) | 1248 households (90%) | 20 000 households (98%) | 4080 individuals (87%) | 4080 individuals (88%) | 1800 households (100%) |
| Sampling strategy | Two-stage cluster randomised sampling with probability proportionate to size, stratified by household wealth category | Population census | Two-stage cluster randomised sampling with probability proportionate to size, stratified by urban and rural | Two-stage cluster randomised sampling with probability proportionate to size, stratified by urban and rural | Two-stage cluster randomised sampling across four purposively selected administration units† |
| Study definition of disability status | Yes/no reported by household head followed by list of conditions | Washington Group Short Set reported by individual if present, or household head if not | Washington Group Short Set+2 (self-report unless<8) or clinical impairment | Washington Group Short Set+2 (self-report unless<8) or clinical impairment | Washington Group extended set (report by household head or primary caregiver) |
| Population-level WASH data | *Yes—sanitation only* | Yes | *No—data for case and control households only* | *No—data for case and control households only* | Yes |
| Household-level WASH data | *Yes—sanitation only* | Yes | *Yes—data for case and control households only* | *Yes—data for case and control households only* | Yes |
| Individual-level WASH data | *Yes—people with disabilities only* | Yes—sanitation only | *Yes—people with disabilities only* | *Yes—people with disabilities only* | *Yes—people with disabilities only* |

*Environmental Health Group at the London School of Hygiene and Tropical Medicine (LSHTM), the Water Engineering and Development Centre at Loughborough University, Mzuzu University and the Centre for Social Research at the University of Malawi.
†Purposive selection criteria: limited exposure to sanitation promotion and relative ease of access.
WASH, water, sanitation and hygiene.
ICED, International Centre for Evidence in Disability

Bangladesh-2, Malawi) or with people with disabilities and matched control subjects (Cameroon, India) as follows:

► Type of sanitation facility;
► Ownership of facility;
► Drinking water source (excluded in Bangladesh-1);
► Time to reach water point, collect water and return to household (excluded in Bangladesh-1);
► Recent episodes of diarrhoea (Malawi: in last 7 days; India/Cameroon: previous 4 weeks; Bangladesh-1 and Bangladesh-2 excluded).

Individual data were not available for the Bangladesh-2 study. Across the remaining four data sets, individual-level WASH access data were collected from all participants identified to have a disability in the survey, and via a proxy (usually their primary caregiver or the head of the household) for those who were not able to communicate independently. The following data were collected from individuals:

► Use of the same sanitation facility as other household members;
► Reasons if a different facility is used;
► Whether the facility can be accessed without contact with faeces;
► Whether drinking water can be collected by the individual;
► Reasons if drinking water cannot be collected by the individual;
► Whether water can be accessed within the household without assistance;
► Whether the same bathing place is used as other household members.

**Other data collected**

In each survey, the household head (Bangladesh-1, Bangladesh-2, Malawi) or participants with disabilities and matched control subjects (Cameroon, India) reported

on the demographic composition of the household and provided data on socioeconomic status (SES) (eg, asset ownership, material of the roof/floor/wall, income).

## Data analysis

Data were analysed in Stata V.12.0.[27] Descriptive statistics were used to describe each dataset.

Disability prevalence estimates for each study area were determined using the svy command to account for clustering in each of the survey designs. Severity of disability was calculated for all datasets except Bangladesh-1 (which did not include a measure of severity) as follows:

► Moderate: those with moderate clinical impairments or reporting maximum 'a lot of difficulty' in at least one domain of the WGSS.
► Severe: those with severe clinical impairments or reporting 'cannot do' in at least one domain of the WGSS.

SES indices were created for Bangladesh-1, Cameroon and India using principal component analysis of assets, and divided into quartiles.[28] Bangladesh-2 data were precoded by interviewers at the data collection stage as ultra-poor, poor and non-poor based on land ownership and livelihood category.[29] In Malawi, annual average household income was collected as a continuous variable and divided into quartiles.

Using the JMP Millennium Development definitions, household water and sanitation facilities were classified as improved or unimproved as follows[7]:

► Improved sanitation: connected to public or private sewer; connected to septic system; pour-flush latrine; simple pit latrine; ventilated improved pit latrine;
► Unimproved sanitation: public or shared latrine; open pit latrine; bucket latrine;
► Improved water source: piped into household; public standpipe; borehole; protected dug well; protected spring; rainwater collection;

► Unimproved water source: unprotected well, unprotected spring; rivers or ponds; vendor-provided water; bottled water; tanker truck water.

We assessed whether household-level access to WASH was different in households with a disabled member compared with households that included no disabled member using multivariable logistic regression. In Malawi and Bangladesh-2, these data were available for all households. In Cameroon and India, these data were available for all households that included a person with a disability, and for age-sex-cluster-matched control households that did not include a person with a disability. Household-level water access data were not available for the Bangladesh-1 study.

We evaluated whether individuals with disabilities experienced poorer WASH access than other members of their household by describing the reported difficulties of people with disabilities in accessing the same WASH facilities as other household members (data not available for Bangladesh-2 study). Predictors (eg, type of impairment, age or sex) of good intrahousehold access to WASH among people with disabilities were explored using multivariable logistic regression for the Cameroon, India and Malawi studies.

Missing data were excluded from analyses and listed as such in relevant tables.

## RESULTS

A total of 99 254 individuals in 24 189 households were sampled across the five datasets (ranging between 3567 and 75 767 per dataset). The age and sex distribution of the datasets are given in table 2. In all, the population was relatively evenly distributed between males and females, and a large proportion of the sample were children or young adults. The household size ranged from an average of 3.9 in Bangladesh-2 to 7.8 in Cameroon.

| Table 2 | Study sample characteristics | | | | |
|---|---|---|---|---|---|
| | **Bangladesh -1** | **Bangladesh-2** | **Cameroon** | **India** | **Malawi** |
| Sample size: households | 1207 | 19 627 | 677 | 875 | 1803 |
| Sample size: individuals | 7511 | 75 767 | 3467 | 3573 | 8834 |
| Sex | | | | | |
| Female | 3867 (51.5%) | 37 076 (48.9%) | 2112 (59.2%) | 1866 (52.2%) | 4500 (50.9%) |
| Age group (years)* | | | | | |
| 0–17 | 2802 (37.3%) | 26 392 (34.8%) | 1950 (54.7%) | 1223 (34.2%) | 4562 (51.6%) |
| 18–24 | 909 (12.1%) | 49 375 (65.2%) | 317 (8.9%) | 473 (13.2%) | 1179 (13.3%) |
| 25–44 | 2090 (27.8%) | – | 572 (16.0%) | 986 (27.6%) | 1862 (21.1%) |
| 45–64 | 1204 (16.0%) | – | 396 (11.1%) | 689 (19.3%) | 826 (9.4%) |
| 65+ | 506 (6.7%) | – | 332 (9.3%) | 202 (5.7%) | 405 (4.6%) |
| Mean household size | 6.2 (SD 3.4) | 3.9 (SD 1.5) | 7.8 (SD 4.8) | 5.2 (SD 2.8) | 4.9 (SD 2.1) |

*Age categories restricted to 0–1, 2–9, 0–1, 2–9, 10–19 and 20+ years in Bangladesh-2 raw data; therefore, Bangladesh-2 age data refer to 0–19 and 20+ years.

**Table 3** Prevalence of disability in the five datasets (%, 95% CI)*

| | Bangladesh-1 | Bangladesh-2 | Cameroon | India | Malawi |
|---|---|---|---|---|---|
| Sample size (individuals) | 7511 | 75 767 | 3467 | 3573 | 8834 |
| Overall prevalence of disability* | 1.3 (1.0 to 1.7) | 1.9 (1.6 to 2.1) | 11.0 (9.4 to 12.7) | 12.6 (10.9 to 14.5) | 2.6 (2.4 to 2.8) |
| WGSS prevalence of disability | – | 1.9 (1.6 to 2.1) | 5.8 (4.6 to 7.4) | 7.1 (5.7 to 8.9) | 2.6 (2.4 to 2.8) |
| Sex | | | | | |
| Male | 1.3 (1.0 to 1.8) | 2.0 (1.7 to 2.3) | 10.6 (8.9 to 12.5) | 12.0 (9.9 to 14.4) | 2.5 (2.1 to 3.1) |
| Female | 1.3 (0.9 to 1.9) | 1.8 (1.5 to 2.0) | 11.2 (9.3 to 13.5) | 13.2 (11.3 to 15.3) | 2.7 (2.3 to 3.1) |
| Age group (years) | | | | | |
| 2–17 | 0.5 (0.3 to 0.9) | 1.1 (0.9 to 1.2) | 4.9 (3.9 to 6.1) | 3.8 (2.7 to 5.3) | 1.2 (1.0 to 1.5) |
| 18–49 | 0.5 (0.3 to 0.9) | 2.3 (1.9 to 2.6)† | 6.9 (5.3 to 9.1) | 8.2 (6.0 to 11.0) | 1.6 (1.3 to 2.1) |
| 50+ | 4.7 (3.4 to 6.5) | – | 33.7 (28.8 to 38.9) | 38.3 (33.6 to 43.3) | 11.5 (10.2 to 12.9) |
| Type of functional difficulty‡ | | | | | |
| Hearing | 9.7 (4.5 to 17.6) | 21.1 (19.0 to 23.4) | 38.3 (33.2 to 43.5) | 39.3 (34.6 to 44.0) | 29.0 (23.1 to 35.6) |
| Visual | 6.5 (2.4 to 13.5) | 26.5 (24.2 to 28.9) | 31.4 (26.6 to 36.5) | 37.8 (33.2 to 42.5) | 39.2 (32.6 to 40.0) |
| Physical | 41.9 (31.8 to 52.6) | 57.6 (55.0 to 60.3) | 53.3 (48.0 to 58.5) | 51.7 (46.9 to 56.5) | 50.2 (43.4 to 57.1) |
| Intellectual | 20.4 (12.8 to 30.1) | 27.4 (25.0 to 29.8) | 20.2 (16.2 to 24.7) | 36.8 (21.2 to 59.0) | 25.3 (19.7 to 31.7) |
| Age-related | 21.5 (13.7 to 31.2) | – | – | – | – |
| Proportion households with at least one person with a disability | 7.1 (5.6 to 9.0) | 6.5 (5.7 to 7.5) | 39.3 (34.0 to 44.7) | 38.5 (34.1 to 43.1) | 10.9 (9.5 to 12.4) |

*Disability prevalence data restricted to population aged 2+ years only.
†Age categories restricted to 0–1, 2–9, 10–19 and 20+ years in Bangladesh-2 raw data; therefore, Bangladesh-2 age data refer to 0–19 and 20+ years.
‡Functional difficulty reported among people classified as having a disability. These figures are not prevalence estimates and the categories are not mutually exclusive.
WGSS, Washington Group Short Set.

Socio economic status did not differ between households with and without a person with a disability in Bangladesh-1, Bangladesh-2, India and Malawi (data not shown). In Cameroon, mean asset scores were significantly lower for households with people with disabilities (− 0.39 , SD 2.4) than among households without people with disabilities (0.37, SD 2.6).

There was a broad range in the prevalence of disability across the settings: 1.3% (95% CI 1.0 to 1.7) in Bangladesh-1; 1.9% (95% CI 1.6 to 2.1) in Bangladesh-2; 2.6% (95% CI 2.4 to 2.8) in Malawi; 11.0% (95% CI 9.4 to 12.7) in Cameroon and 12.6% (95% CI 10.9 to 14.5) in India (table 3). Four studies used iterations of the Washington Group Questions, supplemented by clinical tools in Cameroon and India. Disability prevalence restricted to the WGSS only was similar across settings: 1.9% (95% CI 1.6 to 2.1) in Bangladesh-2; 2.6% (95% CI 2.4 to 2.8) in Malawi; 5.8% (95% CI 4.6 to 7.4) in Cameroon and 7.1% (95% CI 5.7 to 8.9) in India.

In each study, prevalence was similar between men and women, and increased strongly with age. The most frequent type of functional limitation reported was physical (walking or climbing stairs) in each of the five datasets. The proportions of people with disabilities reporting other functional difficulties (hearing, visual or intellectual) varied across the surveys, but were overall broadly similar in four of the five datasets. Bangladesh-1 included 'age-related' as a type of disability, which comprised 21.5% of the distribution.

The proportion of households including one or more persons with disabilities varied substantially between 6.5% in Bangladesh-2 and 39.3% in Cameroon.

In Bangladesh-1 and Bangladesh-2, WASH access was compared between all households either with or without a person with a disability (as identified in the population-based survey). In Cameroon, India and Malawi, comparison was made between households with a person with a disability and households with age-sex-cluster-matched controls only. Table 4 provides the sample characteristics for these nested case-control studies, which are smaller in sample size compared with the two Bangladesh comparative survey samples.

Table 5 presents access to WASH among households with and without persons with disabilities in each dataset. No statistically significant differences in access to an improved sanitation facility were observed between households with and without a person with a disability in any of the five datasets. In Bangladesh-2, households

**Table 4** Nested case-control study sample characteristics in Cameroon, India and Malawi

| | Cameroon | | India | | Malawi | |
|---|---|---|---|---|---|---|
| | No disability | Disability | No disability | Disability | No disability* | Disability |
| Sample size (individuals) | 274 | 429 | 337 | 508 | 200 | 215 |
| | N (%) | N (%) | N (%) | N (%) | N (%) | N (%) |
| Age group (years) | | | | | | |
| 5–17 | 90 (33) | 114 (27) | 49 (15) | 67 (13) | – | 47 (22) |
| 18–49 | 87 (32) | 87 (20) | 160 (48) | 177 (35) | 142 (71) | 54 (25) |
| 50+ | 97 (35) | 228 (53) | 128 (38) | 264 (52) | 58 (29) | 113 (53) |
| Sex | | | | | | |
| Male | 113 (41) | 179 (42) | 163 (48) | 236 (46) | 158 (79) | 101 (47) |
| Female | 161 (59) | 250 (58) | 174 (52) | 272 (54) | 42 (21) | 113 (53) |
| SES quartile | | | | | | |
| First quartile (poorest) | 64 (23) | 104 (24) | 67 (20) | 142 (28) | 29 (15) | 51 (24) |
| Second quartile | 56 (21) | 117 (27) | 87 (26) | 120 (24) | 81 (41) | 79 (37) |
| Third quartile | 61 (22) | 105 (24) | 94 (28) | 124 (24) | 43 (22) | 37 (17) |
| Fourth quartile (richest) | 92 (34) | 103 (24) | 89 (26) | 121 (24) | 47 (24) | 46 (21) |
| Disability type* | | | | | | |
| Hearing | – | 142 (33) | – | 189 (37) | – | 61 (29) |
| Visual | – | 123 (29) | – | 176 (35) | – | 81 (48) |
| Physical | – | 257 (58) | – | 296 (58) | – | 101 (47) |
| Intellectual | – | 119 (28) | – | 38 (7) | – | 51 (24) |

*Household head details.

with a person with a disability were more likely to share their sanitation facility with another household than were households having no member with a disability (OR 1.3, 95% CI 1.1 to 1.5), but there were no other differences in household sanitation facility ownership across datasets.

There was no difference in prevalence of access to an improved water source between households with and those without a member with a disability in any of the datasets. In Bangladesh-2 and Malawi, there was also no observed relationship between the length of time taken to collect water and disability presence in the household. However, households with a person with a disability were more likely to spend >30 min fetching household water than households without a disabled member in both Cameroon (OR 1.9, 95% CI 1.1 to 3.5) and India (OR 2.4, 95% CI 1.2 to 4.8). No differences in frequency of reported diarrhoeal episodes were observed by individual disability status in Cameroon or India (Cameroon OR 1.2, 95% CI 0.8 to 1.9; India OR 0.9, 95% CI 0.6 to 1.4), or household disability status in Malawi (OR 0.7, 95% CI 0.4 to 1.2).

Access to WASH among people with disabilities in comparison to their household members without disabilities is shown in table 6. The majority of people with disabilities reported that they were able to access the same sanitation facility as other members of their household (including those households without access to any facility). Among those who could not, physical access issues were most commonly reported. A substantial proportion of people with disabilities were not able to access sanitation facilities without either themselves or their clothing coming into contact with faeces. Few people with disabilities in Bangladesh-1 were able to collect water from the source themselves (21%), although the sample size was relatively small (n=78), while this proportion was higher in the other surveys (52% in Malawi to 77% in India). Physical barriers and disapproval from others were the most commonly reported reasons for persons with disabilities not collecting water themselves across the datasets. Most people with disabilities reported that they could access drinking water in the house without assistance and could use the same bathing place as other household members.

Table 7 presents the correlates of intrahousehold WASH access among people with disabilities for the three datasets that included these data (Cameroon, India and Malawi). People with more severe disabilities generally had poorer WASH access than people with less severe disabilities. For example, people with more severe disabilities had a much lower likelihood (OR 0.1, 95% CI 0.1 to 0.3) of using the same sanitation facility as other household members, or using the facility without coming into contact with faeces (OR 0.2, 95% CI 0.1 to 0.3) than people with less severe disabilities. Furthermore, older people with disabilities were more likely than children with disabilities to have

**Table 5** Access to WASH among households with and without people with disabilities

| | Bangladesh-1 | | | Bangladesh-2 | | | Cameroon | | | India | | | Malawi | | |
|---|---|---|---|---|---|---|---|---|---|---|---|---|---|---|---|
| | No disability n=1121 | Disability n=86 | OR (95% CI) | No disability n=18364 | Disability n=1284 | OR (95% CI) | No disability n=274 | Disability n=429 | OR (95% CI) | No disability n=337 | Disability n=508 | OR (95% CI) | No disability n=200 | Disability n=215 | OR (95% CI) |
| **Sanitation facility** | | | | | | | | | | | | | | | |
| Unimproved | 34.5% | 31.4% | Ref. | 17.4% | 20.3% | Ref. | 92.4% | 91.4% | Ref. | 23.4% | 23.3% | Ref. | 99.5% | 99.5% | Ref. |
| Improved | 65.5% | 68.6% | 1.1 (0.7 to 1.8) | 82.7% | 79.8% | 0.8 (0.7 to 1.0) | 7.6% | 8.6% | 0.9 (0.5 to 1.6) | 76.6% | 76.7% | 1.0 (0.7 to 1.4) | 0.5% | 0.5% | 1.1 (0.1 to 17.3) |
| **Ownership** | | | | | | | | | | | | | | | |
| Used only by household | 82.2% | 81.4% | Ref. | 68.9% | 63.0% | Ref. | 86.0% | 87% | Ref. | 52.9% | 53.3% | Ref. | – | – | – |
| Shared ownership with other households | 8.7% | 10.5% | 1.2 (0.6 to 2.5) | 30.2% | 35.8% | 1.3 (1.1 to 1.5)* | 14.0% | 13.0% | 0.9 (0.6 to 1.4) | 14.4% | 13.5% | 0.9 (0.6 to 1.4) | – | – | |
| Public | 9.1% | 8.1% | 0.9 (0.4 to 2.0) | 0.1% | 0.1% | 2.6 (0.3 to 21.6) | 0 | 0.2% | – | 32.7% | 33.2% | 1.0 (0.7 to 1.4) | – | – | |
| Do not know | – | – | – | 0.9% | 1.2% | 1.4 (0.8 to 2.4) | 0 | 0 | – | 0 | 0 | – | – | – | |
| **Time to water source†** | | | | | | | | | | | | | | | |
| ≤30 minutes | – | – | – | 0% | 0% | – | 94.2% | 89.6% | Ref. | 96.6% | 92.1% | Ref. | 86.5% | 80.9% | Ref. |
| >30 minutes | – | – | – | 100% | 100% | – | 5.8% | 10.4% | 1.9 (1.1 to 3.5)* | 3.4% | 7.9% | 2.4 (1.2 to 4.8)* | 13.5% | 19.1% | 1.5 (0.9 to 2.6) |
| **Water source** | | | | | | | | | | | | | | | |
| Unimproved | – | – | – | 0% | 0% | – | 39.2% | 37.3% | Ref. | 23.4% | 23.3% | Ref. | 12.5% | 11.6% | Ref. |
| Improved | – | – | – | 100% | 100% | – | 60.8% | 62.7% | 1.1 (0.8 to 1.5) | 76.6% | 76.7% | 1.0 (0.7 to 1.4) | 87.5% | 88.4% | 1.1 (0.6 to 2.0) |
| At least one episode of diarrhoea in the last 4 weeks‡ | – | – | – | – | – | – | 11.7% | 13.8% | 1.2 (0.8 to 1.9) | 12.8% | 11.8% | 0.9 (0.6 to 1.4) | 21.5% | 17.2% | 0.8 (0.5 to 1.3) |

*Statistical significance at the 95% confidence level.
†Round-trip travel time.
‡Refers to any episode of diarrhoea in the household on a 7-day recall-period in Malawi; and any episode of diarrhoea experienced by the individual respondent on a 4-week recall-period in India and Cameroon.
WASH, water, sanitation and hygiene.

**Table 6** Access to WASH among individuals with disabilities

| | Bangladesh -1 | Bangladesh -2 | Cameroon | India | India: open defecation only | Malawi | Malawi: open defecation only |
|---|---|---|---|---|---|---|---|
| | n=78* | n=1374 | n=429 | n=508 | n=274 | n=215 | n=36 |
| Use same sanitation facility as other household members | 81% | 82.0% | 90% | 95% | 94% | 94.4% | 89% |
| Reasons for using a different facility† | | | | | | | |
| Physically impossible | 93% | – | 73% | 28% | 19% | 33% | 43% |
| Others would not like it/not allowed | | | 5% | 0 | 0 | 17% | 0 |
| Physical or verbal abuse | | | 0 | 0 | 0 | 17% | 0 |
| I would be embarrassed | | | 0 | 0 | 0 | 8% | 0 |
| Other | 7% | | 23% | 72% | 81% | 25% | 57% |
| Can access sanitation facility without contact with faeces | 53% | – | 86% | 58% | 55% | 86% | 83% |
| Can collect drinking water themselves | 21% | – | 69% | 77% | – | 53% | 67% |
| Reasons cannot collect drinking water themselves‡ | | | | | | | |
| Physically impossible | 50% | | 80% | 86% | – | 74% | – |
| Others would not like it/not allowed | 50% | | 13% | 5% | | 8% | |
| Physical or verbal abuse | 0 | | 0 | 0 | | 3% | |
| I would be embarrassed | 0 | | 0 | 0 | | 5% | |
| Other | 0 | | 8% | 9% | | 10% | |
| Can access drinking water at home without assistance§ | 88% | – | 86.% | 59% | – | 84% | 81% |
| Use same bathing source as other household members§ | 81% | – | 98% | 97% | – | 95% | 86% |

*Eight households with a confirmed member with a disability excluded due to missing data.
†This is of among the proportion who state they use a different facility.
‡This is of among the proportion who state they cannot collect water themselves.
§This is among the proportion that collect water themselves.
WASH, water, sanitation and hygiene.

access to the same sanitation facilities as other household members in all three settings. There was evidence that among people with disabilities, women were more likely than men to use the same sanitation facilities as other members of the household in Malawi, and to collect water for themselves in Cameroon and Malawi.

## DISCUSSION

Poorer access to WASH among people with disabilities across low-income and middle-income countries is 'widely acknowledged but little studied'.[15] This analysis of five cross-sectional, single or multidistrict, population-based surveys provides some of the first comparable quantitative data on the relationship between disability and WASH at the individual and household level, including multi-country evidence from two African and two South-Asian settings.

No differences in access to improved latrines were observed between households with and without a person with a disability in any of the five datasets. Households with a person with a disability were more likely to have to spend over 30 min (round-trip) fetching water than those without at the Cameroon and Indian sites, and

households including a person with a disability were more likely to use a shared sanitation facility in Bangladesh-2. The former finding may suggest that in these two settings households with one or more people with disabilities live further away from centrally located water points (ie, community peripheries). Second, given that the questions were posed directly to people with disabilities in the house, it may reflect their increased time to complete the same journey as others. Shared sanitation facilities of any description were previously considered unimproved by the JMP, due to concerns of privacy and hygiene.[30] However, the 2017 update acknowledges improved shared facilities, and a recent study by Nelson et al estimated that 37% of households in Bangladesh (similar to proportions identified in Bangladesh-2 data) used a shared but improved facility, and that 75% of households reported satisfaction with the facility.[31 32] This suggests that the use of shared sanitation facilities does not necessarily equate to negative sanitation experience.

Within the household, most people with disabilities used the same sanitation facility as other members of their family, and frequently reported coming into direct contact with faeces when they did so. Furthermore, the

**Table 7** Age-sex-adjusted correlates of WASH access among people with disabilities

| | Same sanitation facility as other household members | | | Access without contact with faeces | | | Collect water for themselves | | |
|---|---|---|---|---|---|---|---|---|---|
| | No n (%) | Yes n (%) | Age-adjusted and sex-adjusted OR (95% CI) | No n (%) | Yes n (%) | Age-adjusted and sex-adjusted OR (95% CI) | No n (%) | Yes n (%) | Age-adjusted and sex-adjusted OR (95% CI) |
| **Cameroon (n=412)** | | | | | | | | | |
| **Age group (years)** | | | | | | | | | |
| 5–17 | 18 (45.0) | 93 (25.0) | Ref. | 24 (40.7) | 87 (24.7) | Ref. | 26 (20.2) | 85 (30.0) | Ref. |
| 18–49 | 5 (12.5) | 81 (21.8) | 2.8 (1.0 to 8.0) | 11 (18.6) | 75 (21.3) | 1.8 (0.8 to 4.0) | 19 (14.7) | 67 (23.7) | 1.0 (0.5 to 1.9) |
| 50+ | 17 (42.5) | 198 (53.2) | 2.1 (1.0 to 4.3) | 24 (40.7) | 191 (54.1) | 2.1 (1.2 to 4.0)* | 84 (65.1) | 131 (46.3) | 0.4 (0.3 to 0.7)* |
| **Sex** | | | | | | | | | |
| Male | 24 (60.0) | 150 (40.3) | Ref. | 29 (49.2) | 145 (41.1) | Ref. | 70 (54.3) | 104 (36.8) | Ref. |
| Female | 16 (40.0) | 222 (59.7) | 2.0 (1.0 to 4.0) | 30 (50.9) | 208 (58.9) | 1.3 (0.7 to 2.3) | 59 (45.7) | 179 (63.3) | 2.1 (1.4 to 3.3)* |
| **Type** | | | | | | | | | |
| Multiple | 23 (57.5) | 126 (33.9) | 0.6 (0.3 to 1.3) | 29 (49.2) | 120 (34.0) | 0.8 (0.4 to 1.5) | 62 (48.1) | 87 (30.7) | 0.6 (0.4 to 1.1) |
| Hearing | 2 (5.0) | 49 (13.2) | 2.3 (0.5 to 10.9) | 5 (8.5) | 46 (13.0) | 1.5 (0.5 to 4.3) | 13 (10.1) | 38 (13.4) | 1.6 (0.8 to 3.5) |
| Seeing | 2 (5.0) | 49 (13.2) | 2.9 (0.6 to 13.8) | 3 (5.1) | 48 (13.6) | 3.2 (0.9 to 11.3) | 12 (9.3) | 39 (13.8) | 1.8 (0.8 to 4.0) |
| Intellectual | 0 (0.0) | 27 (7.3) | – | 0 (0.0) | 27 (7.7) | – | 2 (1.6) | 25 (8.8) | 4.5 (1.0 to 20.7) |
| Physical | 13 (32.5) | 121 (32.5) | Ref. | 22 (37.3) | 112 (31.7) | Ref. | 40 (31.0) | 94 (33.2) | Ref. |
| **Severity** | | | | | | | | | |
| Moderate | 19 (47.5) | 324 (87.1) | Ref. | 32 (54.2) | 311 (88.1) | Ref. | 89 (69.0) | 254 (89.8) | Ref. |
| Severe | 21 (52.5) | 48 (12.9) | 0.1 (0.1 to 0.3)* | 27 (45.8) | 42 (11.9) | 0.2 (0.1 to 0.3)* | 40 (31.0) | 29 (10.3) | 0.2 (0.1 to 0.3)* |
| **India (n=508)** | | | | | | | | | |
| **Age group (years)** | | | | | | | | | |
| 5–17 | 9 (36.0) | 58 (12.0) | Ref. | 28 (13.0) | 39 (13.3) | Ref. | 18 (15.4) | 49 (12.5) | Ref. |
| 18–49 | 11 (44.0) | 166 (34.4) | 2.4 (0.9 to 6.0) | 75 (34.9) | 102 (34.8) | 1.0 (0.6 to 1.7) | 27 (23.1) | 150 (38.4) | 2.0 (1.0 to 4.0) |
| 50+ | 5 (20.0) | 259 (53.6) | 8.2 (2.6 to 25.4)* | 112 (52.1) | 152 (51.9) | 1.0 (0.6 to 1.7) | 72 (61.5) | 192 (49.1) | 1.0 (0.5 to 1.8) |
| **Sex** | | | | | | | | | |
| Male | 10 (40.0) | 231 (47.8) | Ref. | 98 (45.6) | 143 (48.8) | Ref. | 58 (49.6) | 183 (46.8) | Ref. |
| Female | 15 (60.0) | 252 (52.2) | 0.7 (0.3 to 1.6) | 117 (54.4) | 150 (51.2) | 0.9 (0.6 to 1.3) | 59 (50.4) | 208 (53.2) | 1.1 (0.7 to 1.7) |
| **Type** | | | | | | | | | |
| Multiple | 13 (52.0) | 186 (38.5) | 0.8 (0.3 to 2.0) | 91 (42.3) | 108 (36.9) | 0.8 (0.5 to 1.3) | 57 (48.7) | 142 (36.3) | 1.0 (0.6 to 1.8) |

Continued

**Table 7** Continued

| | Same sanitation facility as other household members | | | Access without contact with faeces | | | Collect water for themselves | | |
|---|---|---|---|---|---|---|---|---|---|
| | No n (%) | Yes n (%) | Age-adjusted and sex-adjusted OR (95% CI) | No n (%) | Yes n (%) | Age-adjusted and sex-adjusted OR (95% CI) | No n (%) | Yes n (%) | Age-adjusted and sex-adjusted OR (95% CI) |
| Hearing | 1 (4.0) | 78 (16.2) | 5.4 (0.7 to 42.7) | 26 (12.1) | 53 (18.1) | 1.5 (0.8 to 2.5) | 14 (12.0) | 65 (16.6) | 2.3 (1.1 to 4.6)* |
| Seeing | 0 (0.0) | 76 (15.7) | – | 35 (16.3) | 41 (14.0) | 0.8 (0.4 to 1.4) | 9 (7.7) | 67 (17.1) | 3.4 (1.5 to 7.6)* |
| Intellectual | 1 (4.0) | 16 (3.3) | 0.8 (0.1 to 7.7) | 7 (3.3) | 10 (3.4) | 2.1 (0.4 to 11.0) | 4 (3.4) | 13 (3.3) | 0.9 (0.2 to 5.0) |
| Physical | 10 (40.0) | 127 (26.3) | Ref. | 56 (26.1) | 81 (27.7) | Ref. | 33 (28.2) | 104 (26.6) | Ref. |
| Severity | | | | | | | | | |
| Moderate | 17 (68.0) | 364 (75.4) | Ref. | 150 (69.8) | 231 (78.8) | Ref. | 94 (80.4) | 287 (73.4) | Ref. |
| Severe | 8 (32.0) | 119 (24.6) | 0.9 (0.4 to 2.2) | 65 (30.2) | 62 (21.2) | 0.6 (0.4 to 0.9)* | 23 (19.7) | 104 (26.6) | 1.4 (0.8 to 2.3) |
| **Malawi (n=215)** | | | | | | | | | |
| Age group | | | | | | | | | |
| 2–17 | 6 (50.0) | 40 (19.9) | Ref. | 10 (34.5) | 35 (19.1) | Ref. | 14 (14.0) | 33 (29.0) | Ref. |
| 18–49 | 5 (41.7) | 49 (24.4) | 1.5 (0.4 to 5.2) | 2 (6.9) | 52 (28.4) | 7.4 (1.5 to 36.1)* | 20 (20.0) | 34 (29.8) | 0.7 (0.3 to 1.7) |
| 50+ | 1 (8.3) | 112 (55.7) | 18.2 (2.1 to 157.6)* | 17 (58.6) | 96 (52.5) | 1.6 (0.6 to 3.8) | 66 (66.0) | 47 (41.2) | 0.3 (0.1 to 0.6)* |
| Sex | | | | | | | | | |
| Male | 5 (41.7) | 96 (47.8) | Ref. | 15 (51.7) | 86 (47.0) | Ref. | 52 (50.0) | 49 (43.0) | Ref. |
| Female | 7 (58.3) | 105 (52.4) | 0.6 (0.2 to 2.0) | 14 (48.3) | 97 (53.0) | 1.3 (0.6 to 2.8) | 48 (48.0) | 65 (57.0) | 1.7 (1.0 to 3.1) |
| Type | | | | | | | | | |
| Multiple | 6 (50.0) | 67 (35.1) | Ref. | 16 (55.2) | 56 (32.4) | Ref. | 43 (44.8) | 30 (27.8) | Ref. |
| Hearing | 2 (16.7) | 25 (13.1) | 1.9 (0.3 to 11.0) | 1 (3.5) | 26 (15.0) | 9.7 (1.1 to 82.1)* | 6 (6.3) | 21 (19.4) | 5.0 (1.7 to 14.8)* |
| Seeing | 0 | 42 (22.0) | – | 2 (6.9) | 40 (23.1) | 6.4 (1.4 to 30.4)* | 19 (19.8) | 23 (21.3) | 2.0 (0.9 to 4.6) |
| Intellectual | 1 (8.3) | 13 (6.8) | 2.6 (0.3 to 25.4) | 3 (10.3) | 11 (6.4) | 1.5 (0.3 to 7.1) | 3 (3.1) | 11 (10.2) | 4.2 (1.0 to 17.4) |
| Physical | 3 (25.0) | 44 (23.0) | 1.1 (0.2 to 5.1) | 7 (24.1) | 40 (23.1) | 1.5 (0.6 to 4.2) | 25 (26.0) | 23 (21.3) | 1.5 (0.7 to 3.2) |
| Severity | | | | | | | | | |
| Moderate | 7 (58.3) | 152 (79.6) | Ref. | 14 (48.3) | 145 (83.8) | Ref. | 62 (64.6) | 97 (89.8) | Ref. |
| Severe | 5 (41.7) | 39 (20.4) | 0.4 (0.1 to 1.4) | 15 (51.7) | 28 (16.2) | 0.2 (0.1 to 0.4)* | 34 (35.4) | 11 (10.2) | 0.2 (0.1 to 0.3)* |

*Statistical significance at the 95% confidence level.
WASH, water, sanitation and hygiene.

majority of people with disabilities in Bangladesh and almost half in Malawi were unable to collect drinking water from the source themselves. These analyses also showed that older people generally had better access than children, although given the different WASH practices of children and older people it is difficult to know whether or not this was related to disability.

Overall, few differences in WASH access were observed between households that included a member with a disability and those that did not. However, there is a need to better understand the quality of WASH services accessed by people with disabilities, particularly for those whose households do not have access to improved facilities. In India, for example, almost two-thirds of people with disabilities in the sample practised open defecation, and over half were unable to do this without coming into direct contact with faeces. These findings are of concern given the established association between direct contact with faeces and substantially increased risk of both diarrhoeal disease and hygiene-related stigma.[33] In addition, several recent studies have observed survey respondents' preference for open defecation in rural India even when latrines are available, highlighting the need for clearer and sustained promotion on disability inclusive latrine use in this setting.[34 35]

There was evidence that people with disabilities have more difficulties both collecting water themselves and accessing it within their homes across datasets. Lower volume of water consumption increases dehydration and the potential for increased morbidity.[36] These findings confirm the need for more nuanced quantitative data collection that captures the quality of access to WASH at the individual level and can inform on appropriate WASH programming. This is important because previous research has suggested that WASH programmes may inadvertently exclude people with disabilities through the delivery of improved, but inaccessible facilities, highlighting the need for better quality monitoring data that records whether facilities are accessible or not.[15] Our findings show that data on WASH and disability should be collected at the individual rather than the household level in order to identify the concerns facing people with disabilities in this domain. Specifically, these findings promote the inclusion of intrahousehold access questions in the JMP's recommended questions for household surveys, to better understand and overcome lower quality access to WASH among persons with disabilities within their households.

Furthermore, additional domains may need to be included in WASH scores, focusing on items such as appropriateness and quality of WASH available with respect to people with disabilities. Ideally, surveys on disability should routinely include individual WASH data as it represents a major challenge facing people with disabilities in low-income and middle-income countries.[16]

Several meta-analyses have quantified the implications of improved WASH for exposure pathways for diarrhoeal and infectious diseases, thereby affecting disease burden in 'disability-adjusted life years'.[37 38] However, no previous quantitative studies have explored the implications of disability for access to improved WASH, and in particular the implications for quality of WASH access. A recent qualitative study undertaken in Malawi established that needs and barriers among people with disabilities related to WASH varied markedly between individuals, in relation to impairment type, gender and socioeconomic factors,[16] with much more work needed in this area. Programme implementers and policy makers must ensure that the needs of persons with disabilities are incorporated into programme design in a comprehensive, participatory way that includes intrahousehold WASH access.

There were a number of limitations to the design of the analyses that need to be taken into account. While response rates across datasets were high, data on the characteristics of non-responders were not available. Disability prevalence estimates varied markedly across the studies, largely because of the different methods and disability definitions used across the surveys. In Cameroon and India, a self-reported measure of disability was combined with clinical screening for specific impairments, and therefore the prevalence was higher there than in the Bangladesh-2 and Malawi studies where the self-reported measure was used in isolation. In the Bangladesh-1 study, a binary screen was applied, which may explain the low prevalence of disability in that survey. Presenting prevalence estimates based on the WGSS only, provided similar estimates across settings, although these were higher in the datasets where individuals self-reported (Cameroon and India) compared with those in which a household head reported on behalf of the rest of the household (Bangladesh-2 and Malawi). This finding supports the ongoing dialogue regarding the importance of comparable disability measures for use in population-based surveys the international recommendation for which is the use of tools by the Washington Group on Disability Statistics.[39 40] Furthermore, each survey was conducted at the state or regional level and the estimates derived therefore do not constitute national estimates. Finally, comparative data on indicators of quality access to WASH among controls/people without disabilities were not collected in any of the studies. There were also important strengths. The surveys were relatively large, conducted in different geographic areas, and collected broadly consistent information on WASH.

## CONCLUSION

No relationship was observed between household-level WASH access and the presence of a household member with a disability in five surveys conducted in low- or lower-middle income countries. However, at the individual level, while most people with disabilities could access the same facilities as other members of their household, this often entailed contact with faeces and required assistance from others. In particular, people who reported severe limitations experienced the least equitable intrahousehold

WASH access. Type of limitation reported, age and sex were predictive factors of inequitable access in certain cases. Quality of WASH access is poorer among people with disabilities. This requires programmatic and research attention. Future data collection should capture quality of WASH access in a more nuanced way and collect individual level data from people with and without disabilities to make accurate comparisons. Further programmatic work is needed to improve general access to WASH, and this must be inclusive in nature to ensure that the quality of access is equitable among people with disabilities.

**Author affiliations**
[1]International Centre for Evidence in Disability, London School of Hygiene & Tropical Medicine, London, UK
[2]Environmental Health Group, London School of Hygiene & Tropical Medicine, London, UK
[3]Monitoring[4] chΔnge and IRC-Associate, The Hague, The Netherlands
[4]Centre for Social Research, University of Malawi, Zomba, Malawi
[5]Centre for Tropical Medicine and Global Health, Oxford University, Oxford, UK
[6]International Centre for Diarrhoeal Disease Research, Dhaka, Bangladesh
[7]Resarch and Evaluation Division, Bangladesh Rural Advancement Committee, Dhaka, Bangladesh
[8]World Vision International, Geneva, Switzerland

**Acknowledgements** The authors would like to acknowledge the contribution of their colleague, the late James Milner. His untimely and tragic death during fieldwork in Malawi was a great loss to his friends and colleagues as well as to the scientific research community in Malawi and beyond.

**Contributors** All authors reviewed the final draft of the manuscript and agree with the findings and conclusions. IM was responsible for drafting the paper, conducting the data analysis and overseeing the fieldwork in India and Cameroon. W-PS gave oversight on the data analysis and feedback on all manuscript drafts. KB contributed to the study design of the Bangladesh-2 dataset and reviewed manuscript drafts. JC oversaw data collection for the Malawi dataset. LD contributed to the study design of the Malawi dataset, oversaw data collection for the Malawi dataset and reviewed manuscript drafts. AKH was responsible for the study design of the Bangladesh-1 dataset, oversaw data collection and data management for the Bangladesh-1 dataset and reviewed manuscript drafts. SPJ contributed to the study design of the Bangladesh-2 dataset, oversaw data collection for the Bangladesh-2 dataset and reviewed manuscript drafts. SP was responsible for the study design and data collection on India and Cameroon. MR contributed to the study design of the Bangladesh-2 dataset. MS contributed to the study design of the Bangladesh-2 dataset. HK assisted in drafting the paper, and guided the data analysis, and contributed towards designing the study in India and Cameroon. AB was responsible for the study design of the Malawi dataset, contributed to the design of the Bangladesh-2 dataset, gave oversight on data analysis and feedback on all manuscript drafts, and coordinated access to external datasets. AB also contributed to the design of the data collection tool across all study sites.

**Funding** This research has been funded in part by the Australian Aid (Australian Government Department of Foreign Affairs and Trade) through the Australian Development Research Awards Scheme under an award entitled 'Disability and its impact on safe hygiene and sanitation. Bangladesh-1: funded by The Bill & Melinda Gates Foundation, Global Development Grant Number OPP1102989. Bangladesh-2: funded by Australian Aid (Australian Government Department of Foreign Affairs and Trade), Grant number 66469 India and Cameroon: funded by CBM Germany, LSHTM Grant Number ITCRRH71 Malawi: funded by Australian Aid (Australian Government Department of Foreign Affairs and Trade), Grant number 66469.

**Disclaimer** The views expressed in this publication are those of the authors and not necessarily those of the Commonwealth of Australia. The Commonwealth of Australia accepts no responsibility for any loss, damage or injury resulting from reliance on any of the information or views contained in this publication.

**Competing interests** None declared.

**Patient consent** Obtained.

**Ethics approval** The ethical approval obtained for each study is provided in the online supplementary file 1.

**Provenance and peer review** Not commissioned; externally peer reviewed.

**Data sharing statement** Data for each dataset are available from the respective project leads on request.

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
