## [Reviewer comments · BMJ Open]

ARTICLE DETAILS

TITLE (PROVISIONAL)	Access to water and sanitation amongst people with disabilities: results from cross-sectional surveys in Bangladesh, Cameroon, India and Malawi
AUTHORS	Mactaggart, Islay; Schmidt, Wolf-Peter; Bostoen, Kristof; Chunga, Joseph; Danquah, Lisa; Halder, Amal; Parveen Jolly, Saira; Polack, Sarah; Rahman, Mahfuzar; Snel, Marielle; Kuper, Hannah; Biran, Adam

VERSION 1 – REVIEW

REVIEWER	Marin MacLeod Dalla Lana School of Public Health, University of Toronto, Canada.
REVIEW RETURNED	10-Jan-2017

GENERAL COMMENTS	This study certainly addresses a gap in the quantitative literature with regard to WASH access for persons with disabilities. The multi-country data is interesting and, despite some limitations to comparability across countries, presents a valuable overview of common issues and experiences for persons with disabilities in relation to their access to WASH. I believe the following point should be addressed: The study would provide increased value to policy makers and program planners if the authors offered a set of specific recommendations based on the findings of the study. These recommendations could be organized by country, or in relation to their relevance to water, sanitation or hygiene. The recommendations that are currently presented in the study (page 16-17) are very general in nature, and may elicit only limited response from policy and programmatic institutions. A more specific set of recommendations based on the findings would be helpful.
---

REVIEWER	Robert Bain UNICEF USA
REVIEW RETURNED	04-Feb-2017

GENERAL COMMENTS	This is an important and timely study documenting the challenges faced by people with disabilities in accessing water and sanitation services. It has implications for the monitoring and achievement of the SDGs, including the goals for water and sanitation (SDG 6) and reducing inequalities (SDG 10).
--

	Based on large sub-national surveys in four low- and middle-income countries (five settings) the authors demonstrate clearly that comparing WASH access between households with or without a person with disabilities is insufficient and individual level responses are required. When questions are asked of people with disabilities (or a caregiver) the study shows that although most use the same sanitation facility as other members of their households, many cannot do so without contact with faeces. The authors include multivariate analysis to explore the factors associated with the challenges faced by people with disabilities. Four main comments: (1) Was information collected on whether households shared sanitation facilities and whether these were located off premises? These may be important (physical and social) barriers to access by people with disabilities and at least the first of these is part of the JMP core questions document. If these questions were asked then this should be included in the analysis. (2) Different definitions of disability were used in the studies (particularly Bangladesh-1) but most had included the Washington short set. Could you include a sensitivity analysis to see how differently individuals in the other four study settings would not be classified as having a disability? (3) I do not think the data provide enough support for the suggestion that women with disabilities are more likely to use the same sanitation facilities as other household members. (4) The WASH and disability sectors have established "core" questions from the JMP and Washington group respectively but the questions asked to individuals with disabilities are not widely used. Please include the specific formulation of these questions as SI and reflect on their performance across the settings. Which would be most important for the disability and WASH sectors to incorporate in monitoring? For minor comments please refer to the commented pdf (attached). - The reviewer provide a marked copy with additional comments. Please contact the publisher for full details
--	---

REVIEWER	Samantha Winter Rutgers University School of Social Work New Brunswick, NJ USA
REVIEW RETURNED	13-Feb-2017

GENERAL COMMENTS	Major Comments 1) The authors explore the association between disability and access to WASH in middle and low income countries. This is an very important and understudied area. This is the first multi-country, quantitative study addressing these issues and, therefore, it is an important piece. 2) The discussion is well-written and ties in the findings of the study well. The introduction, methods, and results could use some work. 3) In general, there is a need for better definitions for disability and WASH access. In addition, the sample is an issue. It changes multiple times due to list-wise deletion, different units of analysis (individual and household), and different types of analyses. The changes are not well-defined and not well-described. The biggest
--

issue is that the household sample is never clearly defined; yet, it is the basis for several of the analyses. Some of the numbers for the household sample are not adding up properly (please see detailed comments). Finally, the analysis strategy section does not fully describe the analyses conducted. For example, means-testing is used to look at the association between disability and SES. Not only was this not clearly stated in the analysis strategy, but it does not fit in well with the introduction and research questions. While there may be a relationship between WASH access and SES, the authors never made the link. So I am not sure why SES is even treated as an outcome variable instead of a covariate.

Detailed Comments

- 1) Page 4, Line 17; Page 4, Lines 36-41; Page 5, Lines 28-43 : Please clarify the definition of disability being used in this study. It is difficult to determine the definition of disability based on the Introduction and Methods sections. From the description provided, it is very hard to tell what was measured, how it was measured, and how you are utilizing the information from the surveys. Perhaps consider mentioning the 6 core domains in the Methods section or in Table 1 rather than just providing examples (such as hearing, seeing). In addition adding at least a broad definition of disability in the introduction, such as that used by the Washington Group, would help readers who are not familiar with how disability is defined, screened for, or measured (e.g. "For the purpose of determining disability status using census data, persons with disabilities are defined as those who are at greater risk than the general population of experiencing limitations in performing specific tasks (activities) or restrictions of participation in society"). Also, for the "binary screen" used in the Bangladesh-1, providing the actual question that was asked and/or describing the "follow-up questions" would be helpful.
- 2) Page 5, Lines 5-7: Do you have any justification for choosing to look at these particular countries instead of a different sub-set of countries or individual countries? What is the relationship between the countries or did you just choose them because they were the only datasets with comparable disability information. In the discussion you mention low and middle income countries and you also mention having 2 countries from SSA and 3 countries from SE Asia. I know that these are areas of the world that where the lack of sanitation has been most persistent; however, none of that is stated anywhere in the paper. Some justification would be appreciated.
- 3) Page 7, Lines 25-29: Table 1 - You talk about the definitions of disability based on the different measures used across surveys, but they hardly make sense given the lack of detail in the description of the measures. For example, there is no mention of the number of questions/core domains in the Washington Group Short Set, which makes it very difficult to interpret things like "unless <8." Please clarify by adding a better description of the measures in the text or in the table
- 4) Page 5; Lines 48-end of page : Please clarify your WASH measures. You simply tell us who answered the questions/at what level the questions were asked. You cite the WHO Core Questions on Drinking Water and Sanitation, but which items, in particular, did you use? I imagine you only used 3 or 5 of the items in the WHO core Questions to develop your definition of WASH for this study. Please state them. I know word limits can be an issue, but the definition of the measures you are using, e.g. "access to sanitation and water" is critical.
- 5) Page 5, Line 56-57: Please provide a better description of what Bangladesh-2 used. Page 8, Lines 9-13: How can you measure

severity of disability from a binary measure of disability (see p.5, Lines 39-43 for Bangladesh-1 definition of disability measure)?

6) Page 8, Lines 39-40: Was list-wise deletion a justifiable option to deal with missingness? What percent of the data was missing on the independent and dependent variables and covariates? Please provide some information to justify use of list-wise deletion. Given the nature of your analyses (e.g. many different analyses) it seems that list-wise deletion is not a good idea. It is very difficult to follow your samples throughout the different analyses. You often don't provide and/or repeat your different sample sizes. You do not distinguish well between the household versus individual analyses/findings, and, on top of that you have deleted cases for missing that change per analysis. This is very hard for a reader to interpret. I suggest using a single sample size (individual and household) for all analyses and clearly stating what that sample is (individual and household). Alternatively, you need to provide a sample chart that shows what samples are being used for what analyses.

7) Page 9, Lines 45-47; Page 11, Table 4: It seems like SES being treated as a dependent variable in this analysis instead of a covariate. If you are treating it as a dependent variable that needs to be introduced much earlier in the paper, e.g. in the research questions. An argument can be made that SES and access to WASH are linked, but you have not provided any reason as to why you are suddenly treating SES as an outcome variable. Up until page 9, it is implied that SES is a covariate in the models. Also there is no discussion of the analysis strategy for looking at associations between disability and SES in the Data Analysis section. All of a sudden the results appear in the results sections on page 9, Lines 45-47 and in Table 4.

8) Page 9, Lines 55-57: Editing issue with the statement "without people without disabilities"

9) Page 11: Table 4 needs some work. It is difficult to read, largely because SES is measured differently across countries, it is treated as a dependent variable instead of a covariate, and the results are presented differently. The table is also missing information that can help the reader to interpret the findings more quickly. For example, denoting significance would be very helpful for the regression results (ORs) with a footnote at the bottom of the table showing how it is denoted?

10) Page 10, Table 3: Please consider providing restating the sample size of the individuals (as you did in Table 2) in Table 3. Also, you need to provide the total sample size of the households in Table 3 or somewhere. As far as I can tell, you do not provide the sample size of the households anywhere in the paper except, possibly, in Table 4 on Page 11, Lines 7-9, but these samples do not match the percentages of households with at least one person with a disability you provide in Table 3 on Page 10, Lines 26-29.

11) Page 11; Lines 7-9: As mentioned previously, your household percentages do not match between Tables 3 (Page 10, Lines 26-29) and 4 (Page 11, Lines 7-9). According to the figures in Table 3, 7.1%, 6.5%, 39.3%, 38.5%, and 10.9% of households in BDG-1, BDG-2, Cameroon, India, and Malawi, respectively, have at least 1 person with a disability. Then, in Table 4, the "household" sample sizes provided (Page 11, Lines 7-9) imply that the percentages of households with disabilities in the sample are about 7%, 12%, 61%, 60%, and 50%, for BDG-1, BDG-2, Cameroon, India, and Malawi, respectively. This could be a simple issue of you not providing enough information about the household sample sizes used in this analysis or it could be an analysis error. It is difficult to tell without

	more information, but please review and determine how best to let the reader know what is going on here. 12) Page 12, Line 25: Your sample sizes keep changing in these analyses. I assume you are trying to maximize your sample for each analysis, but this changing of sample sizes between analyses makes this very difficult to interpret. I would suggest choosing a single sample size for all analyses or use imputation if it is justifiable. 13) Page 12, Lines 49-end of page : You report that the data is not shown. From where did this information come? Follow up verbatim questions? None of this is discussed in the measures section either. Again, I recognize word limit challenges, but I think it would be appropriate to specify the measure(s) from which these findings were taken and/or how the percentages were assessed. Since this information is actually the focus of your discussion and conclusion, it seems like the data should, in fact, be shown. 14) Table 6, Page 14: You are missing sample information for Malawi. Also, again, your sample sizes are changing and you have no footnote to explain it like in Table 5. Please consider a single list-wise sample or imputation. Also please consider denoting significance by including p-values or notation to make this table faster to interpret for the reader. 15) Page 4, Lines 40-41: I am not entirely sure you answered your third research question. I assume the analyses leading up to and the results in Tables 5 & 6 are the attempt to answer the third research question (factors associated with access to improved WASH among people with disabilities), but I don't see how the outcome variables reflect the definition of improved water/sanitation (with the exception of "access without contact with feces). Perhaps consider revising research question 3.
--	---

REVIEWER	Kristen Heitzinger Kentucky Department for Public Health, USA
REVIEW RETURNED	20-Feb-2017

GENERAL COMMENTS	This is an interesting article written on an important topic. However, major revisions are needed in order for this manuscript to be ready for publication. Specific comments: Page 1, Line 38: Please correct affiliation 6 from "icdr, b" to "icdr,b" Page 2, Line 23: The authors state that there was no difference in "episodes of diarrhoea" when (4-week or 1-week) prevalence of diarrhea (not numbers of episodes) was actually compared. Page 2, Line 17: Please consider replacing "through" with "using" for clarity. Page 3, Line 11: Please do not capitalize "state" or "regional". Page 4, Line 26: I suggest softening the language from "confirmed by qualitative research" to "supported by qualitative research" because this is more defensible statement given the paucity of research on this topic. Page 4, Line 35: I suggest removing the word "both" because three things (disability, access to WASH, and experience of WASH) are
---

noted here.

Page 4, Lines 36-37: The phrasing is confusing because the household, not the person, is the unit of analysis for this comparison. I would rephrase to “1) whether households containing people with disabilities have different access to WASH compared to households without disabled members” or something similar.

Page 5: Please clarify the details of the methods. For example, the authors state the intention to assess “access to WASH” and “improved WASH,” but do not define these terms in the text. Is “access to WASH” defined as having access to both improved water and sanitation? Moreover, there very little discussion of hygiene, thus the outcomes may be more appropriately described as “access to water and sanitation” or “improved water and sanitation.”

Page 5, Line 6: I suggest rephrasing from “one in each of Cameroon” to “one each in Cameroon” for clarity.

Page 5, Lines 13-14: Please consider rephrasing from “One of the studies conducted a census” to “One of the studies collected census data to facilitate random sampling of participants” or something similar. In its present form, the connection between the first phrase of this sentence and the second phrase is not very clear.

Page 5, Lines 17-18: Please consider rephrasing from “people identified in the survey to have disabilities” to “people identified to have disabilities in the survey”. The phrase is clearer when the descriptor is closer to the noun it describes.

Page 5, Line 23: Please specify what boards provided ethical approval for this study so that readers can assess whether the necessary approvals were obtained.

Page 5, Lines 52-57: Please provide more detail regarding who participated in the surveys. For example, who were the “individual members of the household” who participated in the surveys in Cameroon and India and how were they selected? In line 56, who was the proxy for participants who could not communicate independently? Was it the primary caregiver for that person? This information may be helpful to readers in assessing whether the data collected regarding these individuals is likely to be accurate.

Page 6, “Other data collected” section: Who were the individual members of the household who were surveyed in Cameroon and India? Did all individual members present at the time of the visit respond or was a household representative (i.e. the oldest member) selected as the respondent? Please provide more detail. Additionally, for consistency, I suggest that the heading for this section should either be formatted in the same way as for the “Disability assessment” and “WASH assessment” sections or it could be placed in the paragraph above the “Disability assessment” section.

Table 1 comments:

-The response rates are not included for the Bangladesh-2 and Malawi studies. Please include them or indicate that the data are not available.

-In the “sampling strategy” row, “Cross-sectional study”, as described for Bangladesh-2 and Malawi, is a study design, not a sampling strategy. Please specify what sampling strategies were used for these surveys.

-Please write out abbreviations in footnotes (i.e. WG, EHG, LSHTM, WEDC) to improve clarity.

Page 8, Lines 5-8: Please specify if any nonresponse data were available. Given the association between age and disability, if information is available on the ages of the members of households that did not participate in the surveys, this information could be used in the svy analysis to improve the accuracy of the disability prevalence estimates.

Page 8, Line 17: Please provide more detail regarding the coding of SES for the Bangladesh-2 survey to enable replication of methods. For example, what data were used to categorize the SES of the households? Were the households grouped into tertiles?

Page 8, Line 18: Was SES considered as a continuous or categorical variable for the households surveyed in Malawi? By “average household income”, was “average annual household income” intended? Please provide more detail regarding how SES was measured for these households.

Page 8, Lines 27-28: This language is unclear. Please consider rephrasing from “a control household selected through matching an individual by age-sex-cluster to the person with a disability” to “control households selected by age-sex-cluster matching to a household member who did not have a disability” or something similar.

Page 8, Line 32: I suggest replacing “considered” with “tested” or “evaluated” or a similar term because this is what occurred.

Page 8, Line 36: I suggest replacing “calculated” with “identified” because the predictors (variables) were not calculated.

Page 9, Line 10: The statement that the “majority of the population were children or young adults” seems difficult to support because of the broad age categories used in the BGD-2 study, which makes up over 75% of the study population. Please consider modifying this statement in order to better reflect the data.

Page 9, Table 2: Please provide a footnote for each of the BGD-2 estimates specifying the age range of the estimate.

Page 9, Lines 38-41: It seems this section refers to the types of impairment data in Table 3, but it is unclear whether these lines accurately describe the Table 3 data. The Table 3 footnote b, “Type as proportion of people with disabilities amongst those reported to have disabilities, neither prevalence estimate nor mutually exclusive” is very difficult to understand and I suggest rephrasing this. Does this table describe the proportion of disability types among the total number of disabilities (i.e. the total number of disabilities—not people with disabilities—is the denominator)? If so, please consider replacing this footnote with “Proportion of type of impairment out of total number of disabilities” or something similar. Taking this into consideration in the Page 9 text, the units of analysis are disabilities,

not people, and therefore the phrase “accounting for over half of the people identified to have a disability” should be rephrased to “accounting for over half of the disabilities identified” or something similar.

Page 9, Line 43: Please define “age-related” impairment in the methods. If there is frequent overlap between “age-related” impairments with other types of impairments, the authors could consider excluding this category, which would increase the comparability of the results with the other studies.

Page 10, Table 3 comments:

-The inclusion of “(%)” and “proportion” in the title is redundant. Please simplify.

-Given the title, it is unnecessary to include “%” in the bottom row of the table. Please remove.

-Footnote a: Please specify the units of age (i.e. years).

-Footnote b: Please see comment above about clarifying this.

-Footnote **: Please specify what age range was included to generate each estimate.

Page 11, Table 4 comments:

-The column title “Effect size (OR or difference)” makes it very unclear what result is being shown. I suggest being consistent in providing only ORs or indicating with a footnote which estimates are differences.

-Please include an effect size for the “socioeconomic status” row.

-Having no toilet is classified by the WHO/UNICEF Joint Monitoring Programme (JMP) as unimproved sanitation. Please combine these groups or provide justification for why they should be separated. I suggest combining these groups on the basis that this will also simplify interpretation of the results.

-Footnote b is unclear. Is “Return” intended to mean “round-trip travel time”? If so, this should be specified.

-The JMP considers access to an improved water source as being within 30 minutes of total travel time. Please consider combining the 15-30 min. category with the less than 15min. category if, per footnote b, these times represent round-trip travel times.

-Please replace “water facility” with “water source.”

-The estimate of the effect size for the association between the water source and presence of a household member with a disability (OR=1.1, 95% CI=1.2-1.9) is implausible because the effect size lies outside of the 95% confidence interval. Please verify and correct.

-Regarding the row “At least one episode of diarrhoea in the last four weeks,” does this refer to any member of the household? If so, please specify this in footnote c or in the methods section.

-In footnote a, please provide more detail regarding whether average annual household income was used or some other measure.

-In footnote c, please provide a more complete explanation of the differences in data collection by site. For example, "A 7-day recall period was used in Malawi." It is unclear what is intended by "individual not household in Cameroon/India." Please clarify.

Page 12, Lines 5-6: This statement is directly contradicted by the results shown in Table 4, because in Cameroon, households with a disabled member were significantly more likely to have an improved water source. Please modify this statement to accurately reflect the study findings.

Page 12, Line 7: Please correct the spelling of "Bangaldesh-2".

Page 12, Line 7-8: This statement is contradicted by Table 4 because in BGD-2, a time to water source of 15-30min. was significantly associated with a reduced likelihood of being a household with a disabled member. Please modify this statement and/or provide data that demonstrate this statement is true.

Page 12, Lines 12-13: The statement that there were no differences in access to an improved water source seems to repeat the first sentence in this paragraph that was not supported by the data in Table 4. Please consider deleting.

Page 12, Lines 14-17: If the Table 4 row "at least one episode of diarrhoea in the last four weeks" is accurate, then what is being measured is household prevalence of diarrhea. Please rephrase "no differences in the frequency of reported diarrhoeal episodes" to "no differences in prevalence of reported household diarrhoea" or something similar to clarify this point.

Page 12, Table 5: I suggest modifying "Can access drinking water without assistance" to "Can access drinking water in the home without assistance" to improve clarity.

Page 12, Table 5: Please add a "%" to the BGD-2 estimate for consistency with the formatting of the other cells.

Page 13, Lines 3-6: This statement ("Almost all people with disabilities reported...") seems to be contradicted by the data shown in Table 5. Per Table 5, in India, only 58.7% of individuals could access drinking water in the home without assistance. In BGD-1, only 80.8% of individuals could use the same bathing source as other household members. It is inaccurate to represent these prevalences as "almost all people" and therefore I recommend modifying this statement.

Page 13, Line 7: The statement that open defecation "did not affect relative intra-household WASH access" does not appear to be supported by the data shown in Table 5, because intra-household comparisons were not made. Please consider removing this statement or modifying it in order to be better supported by the data.

Page 13, Line 9: Please modify "intra-household access" to "intra-household WASH access" for clarity.

Page 13, Line 10: Please consider modifying the phrase "the three

datasets that collected these data” because a dataset cannot collect data.

Page 13, Lines 15-16: It is unclear on what basis the authors conclude that women with disabilities were “generally more likely...to have the same WASH access as other members of the household.” None of the associations between female gender and use of the same facility as other household members are significant for Cameroon, India or Malawi, and one of the associations is in the opposite direction of the other two. Please remove this sentence or modify it so that it is more clearly supported by the data.

Page 14, Table 6: By the heading “same facility as other household members”, please specify whether this refers to a sanitation facility or water source.

Page 16, Lines 11-12: The statement that “no differences in access to improved...water sources were observed between households with and without a person with a disability” may be contradicted by the estimate for Cameroon provided in Table 4. Please verify this estimate and correct this sentence accordingly.

Page 16, Lines 14-15: It is unclear what data the authors used to support the conclusion that “households with a person with a disability were more likely to have to spend over thirty minutes fetching water than those without at the Cameroonian and Indian sites.” Only the estimate for the Indian site appears to be statistically significant (does not include 1.0) and if the basis for this conclusion is the observation of a positive association, regardless of statistical significance, then the same conclusion should be drawn for the Malawi site.

Page 16, Lines 15-16: The statement that households with people with disabilities live further away from water sources appears not to be supported by the data shown in Table 4, as disability was associated with having a >30min. trip to a water source at only one site. Please provide additional data to support this statement or remove it.

Page 16, Lines 24-28: It is unclear what evidence the authors used to support the statement that “people with more severe limitations generally had less equitable access...” In Table 6, severe disability was significantly associated with a lower likelihood of using the same facility as other household members in only one of the three countries where these data were collected. Including the definition of “WASH access” in the methods section would be helpful to clarify how this conclusion was drawn.

Page 16, Lines 34-35: It is unclear how the authors arrived at the conclusion that “In India...two thirds of households with a disabled member practiced open defecation...”. This does not follow from the 60.5% prevalence of having “no toilet” among households noted in Table 4. Using the data shown in Table 5 ($(274/(274+508))=65\%$), one would draw the conclusion that 65% of individuals with disabilities living in India practice open defecation, but this refers to individuals, not households, and 65% is more accurately described as “nearly two thirds.” Please modify this sentence to more accurately reflect the data shown in the results section.

Page 16, Line 37: Reference 25 (Murray and Lopez) does not

support the authors' statement that "contact with faeces...may put them at substantial risk of diarrhoeal disease and hygiene-related stigma." Moreover, there are more up-to-date estimates of the global burden of disease attributable to WASH. Please cite this literature or other literature that supports this statement.

Page 16, Line 38: The data do not support the authors' statement that "there was evidence that people with disabilities have more difficulties...accessing [water] within their homes" because the access of water within the home was not compared between individuals with and without disabilities in Table 5. Please clarify what evidence the authors used to support this conclusion.

Page 16, Line 42: Reference 26 (Manz 2005) discusses the long-term health effects of mild dehydration. It does not support the authors' statement that difficulty in accessing water in the home is "linked to lower water consumption, dehydration, and the potential for increased morbidity." Please provide additional support for this statement or consider removing it.

Page 16, Lines 49-51: This sentence is unclear, and there is more up-to-date literature describing the effect of improved WASH on the global burden of disease. Please cite this literature (i.e. the 2010 global burden of disease study) and consider rephrasing this sentence as "Several meta-analyses have quantified the effect of improved WASH on the global burden of disease measured in Disability-Adjusted Life Years (DALYs)" or something similar.

Page 16, Line 54: What is intended by the term "quality of WASH access"? Please provide more detail on the aspects of WASH access intended by this term.

Page 16, Lines 57-58: As per my earlier comment, there does not seem to be data to support the conclusion that women with disabilities have better WASH access. Please provide evidence for this conclusion.

Page 17, Line 10: The authors describe the binary measure used in the Bangladesh-1 study as "restrictive," yet the measures used in the other surveys are also binary. Please justify the description of this measure as "restrictive" or consider removing this word from the text.

Page 17, Lines 10-11: The data provided do not support the conclusion that the measure of disability used in the Bangladesh-1 study was the reason that the prevalence was lower than in other studies. Please cite literature supporting this conclusion or modify this sentence to read "which may explain the low prevalence" instead of "explaining the low prevalence."

Page 17, Lines 16-20: This sentence is unclear because the "similar trends" were not specified and no evidence was provided to support the idea that differences in the measurement of disability would affect the assessment of disability severity. Please clarify what is meant by "similar trends" and cite support from the literature to support this statement.

Page 17, Line 21: Please do not capitalize "state" and "regional" because they are not proper nouns.

	Page 17, Line 24: Please include examples of what aspects of WASH would be considered “indicators of quality access to WASH” to clarify this concept. Page 17, Line 36-38: As noted in the methods sections, disability was not consistently defined as a reported measure, but also was defined using clinical criteria in Cameroon and India. Please consider revising this sentence accordingly by removing the word “reported.” Page 17, Line 39: Per my comment above, please provide evidence that female gender was predictive of intra-household WASH access. Page 17, Lines 39-41: The statement that “the quality of WASH access is likely to have been poorer among people with disabilities” is not well supported by the data provided in this manuscript because individuals with and without disabilities were not directly compared. Please consider rephrasing this to read “the quality of WASH access may have been poorer among people with disabilities”.
--	--

VERSION 1 – AUTHOR RESPONSE

Reviewer: 1

Reviewer Name: Marin MacLeod

Institution and Country: Dalla Lana School of Public Health, University of Toronto, Canada.

Competing Interests: None declared.

This study certainly addresses a gap in the quantitative literature with regard to WASH access for persons with disabilities. The multi-country data is interesting and, despite some limitations to comparability across countries, presents a valuable overview of common issues and experiences for persons with disabilities in relation to their access to WASH.

Thank you for your positive feedback.

I believe the following point should be addressed:

The study would provide increased value to policy makers and program planners if the authors offered a set of specific recommendations based on the findings of the study. These recommendations could be organized by country, or in relation to their relevance to water, sanitation or hygiene. The recommendations that are currently presented in the study (page 16-17) are very general in nature, and may elicit only limited response from policy and programmatic institutions. A more specific set of recommendations based on the findings would be helpful.

More specific recommendations have now been included in the discussion.

Reviewer: 2

Reviewer Name: Robert Bain

Institution and Country: UNICEF, USA

Competing Interests: None declared

This is an important and timely study documenting the challenges faced by people with disabilities in accessing water and sanitation services. It has implications for the monitoring and achievement of the SDGs, including the goals for water and sanitation (SDG 6) and reducing inequalities (SDG 10).

Based on large sub-national surveys in four low- and middle-income countries (five settings) the authors demonstrate clearly that comparing WASH access between households with or without a person with disabilities is insufficient and individual level responses are required. When questions are asked of people with disabilities (or a caregiver) the study shows that although most use the same

sanitation facility as other members of their households, many cannot do so without contact with faeces. The authors include multivariate analysis to explore the factors associated with the challenges faced by people with disabilities.

Thank you for your positive feedback.

Four main comments:

(1) Was information collected on whether households shared sanitation facilities and whether these were located off premises? These may be important (physical and social) barriers to access by people with disabilities and at least the first of these is part of the JMP core questions document. If these questions were asked then this should be included in the analysis.

Information on whether on the latrine facility is used by the household independently, shared or public are available for all datasets except Malawi. These data have now been included in table 5.

(2) Different definitions of disability were used in the studies (particularly Bangladesh-1) but most had included the Washington short set. Could you include a sensitivity analysis to see how differently individuals in the other four study settings would not be classified as having a disability?

A prevalence estimate using the WG SS is now provided in Table 3 for each of the four datasets including interations of the Washington Group Questions. This is also discussed in the discussion (Page 20, lines 74 - 81).

(3) I do not think the data provide enough support for the suggestion that women with disabilities are more likely to use the same sanitation facilities as other household members.

We have revised the text and have down-played the potential association between gender and access to sanitation facilities (page 20, line 60 - 65).

(4) The WASH and disability sectors have established "core" questions from the JMP and Washington group respectively but the questions asked to individuals with disabilities are not widely used. Please include the specific formulation of these questions as SI and reflect on their performance across the settings. Which would be most important for the disability and WASH sectors to incorporate in monitoring?

The questions asked in each site are now included as appendix 1. A recommendation on the important questions to include in monitoring has been added to the discussion (page 20, line 52 - 55).

For minor comments please refer to the commented pdf (attached).

Thank you for your detailed comments. These have been addressed in the document.

Reviewer: 3

Reviewer Name: Samantha Winter

Institution and Country: Rutgers University School of Social Work, New Brunswick, NJ USA

Competing Interests: None

Major Comments

1) The authors explore the association between disability and access to WASH in middle and low income countries. This is a very important and understudied area. This is the first multi-country, quantitative study addressing these issues and, therefore, it is an important piece.

Thank you for your positive comments.

2) The discussion is well-written and ties in the findings of the study well. The introduction, methods, and results could use some work.

Thank you for your comments. We have revised the introduction, methods and results in line with the comments from the three reviewers.

3) In general, there is a need for better definitions for disability and WASH access. In addition, the sample is an issue. It changes multiple times due to list-wise deletion, different units of analysis (individual and household), and different types of analyses. The changes are not well-defined and not well-described. The biggest issue is that the household sample is never clearly defined; yet, it is the basis for several of the analyses. Some of the numbers for the household sample are not adding up properly (please see detailed comments). Finally, the analysis strategy section does not fully describe the analyses conducted. For example, means-testing is used to look at the association between disability and SES. Not only was this not clearly stated in the analysis strategy, but it does not fit in well with the introduction and research questions. While there may be a relationship between WASH access and SES, the authors never made the link. So I am not sure why SES is even treated as an outcome variable instead of a covariate.

The sample sizes have been clarified to show the total number of households and individuals in each of the population-based surveys, and the sample characteristics of the nested case-control studies in Cameroon, India and Malawi. SES has been removed as an outcome variable in the results and the analysis section has been updated to ensure all analyses are described.

Detailed Comments

1) Page 4, Line 17; Page 4, Lines 36-41; Page 5, Lines 28-43 : Please clarify the definition of disability being used in this study. It is difficult to determine the definition of disability based on the Introduction and Methods sections. From the description provided, it is very hard to tell what was measured, how it was measured, and how you are utilizing the information from the surveys. Perhaps consider mentioning the 6 core domains in the Methods section or in Table 1 rather than just providing examples (such as hearing, seeing). In addition adding at least a broad definition of disability in the introduction, such as that used by the Washington Group, would help readers who are not familiar with how disability is defined, screened for, or measured (e.g. "For the purpose of determining disability status using census data, persons with disabilities are defined as those who are at greater risk than the general population of experiencing limitations in performing specific tasks (activities) or restrictions of participation in society"). Also, for the "binary screen" used in the Bangladesh-1, providing the actual question that was asked and/or describing the "follow-up questions" would be helpful.

The question sets used in each study have been appended to the manuscript. A definition of disability has been provided in the introduction.

2) Page 5, Lines 5-7: Do you have any justification for choosing to look at these particular countries instead of a different sub-set of countries or individual countries? What is the relationship between the countries or did you just choose them because they were the only datasets with comparable disability information. In the discussion you mention low and middle income countries and you also mention having 2 countries from SSA and 3 countries from SE Asia. I know that these are areas of the world that where the lack of sanitation has been most persistent; however, none of that is stated anywhere in the paper. Some justification would be appreciated.

A rationale for inclusion of the five datasets, and relevance to global WASH challenges, has been added to the introduction (page 4, line 116 - 120).

3) Page 7, Lines 25-29: Table 1 - You talk about the definitions of disability based on the different measures used across surveys, but they hardly make sense given the lack of detail in the description of the measures. For example, there is no mention of the number of questions/core domains in the Washington Group Short Set, which makes it very difficult to interpret things like "unless <8." Please clarify by adding a better description of the measures in the text or in the table

This has been addressed in the methods and via the inclusion of the question sets in the appendix

4) Page 5; Lines 48-end of page : Please clarify your WASH measures. You simply tell us who answered the questions/at what level the questions were asked. You cite the WHO Core Questions on Drinking Water and Sanitation, but which items, in particular, did you use? I imagine you only used

3 or 5 of the items in the WHO core Questions to develop your definition of WASH for this study. Please state them. I know word limits can be an issue, but the definition of the measures you are using, e.g. "access to sanitation and water" is critical.

This has been addressed in the methods and via the inclusion of the question sets in the appendix

5) Page 5, Line 56-57: Please provide a better description of what Bangladesh-2 used. Page 8, Lines 9-13: How can you measure severity of disability from a binary measure of disability (see p.5, Lines 39-43 for Bangladesh-1 definition of disability measure)?

This has been addressed in the methods and via the inclusion of the question sets in the appendix

6) Page 8, Lines 39-40: Was list-wise deletion a justifiable option to deal with missingness? What percent of the data was missing on the independent and dependent variables and covariates? Please provide some information to justify use of list-wise deletion. Given the nature of your analyses (e.g. many different analyses) it seems that list-wise deletion is not a good idea. It is very difficult to follow your samples throughout the different analyses. You often don't provide and/or repeat your different sample sizes. You do not distinguish well between the household versus individual analyses/findings, and, on top of that you have deleted cases for missing that change per analysis. This is very hard for a reader to interpret. I suggest using a single sample size (individual and household) for all analyses and clearly stating what that sample is (individual and household). Alternatively, you need to provide a sample chart that shows what samples are being used for what analyses.

This has been clarified in each table and the same dataset used throughout analyses.

7) Page 9, Lines 45-47; Page 11, Table 4: It seems like SES being treated as a dependent variable in this analysis instead of a covariate. If you are treating it as a dependent variable that needs to be introduced much earlier in the paper, e.g. in the research questions. An argument can be made that SES and access to WASH are linked, but you have not provided any reason as to why you are suddenly treating SES as an outcome variable. Up until page 9, it is implied that SES is a covariate in the models. Also there is no discussion of the analysis strategy for looking at associations between disability and SES in the Data Analysis section. All of a sudden the results appear in the results sections on page 9, Lines 45-47 and in Table 4.

SES has been removed as an outcome variable.

8) Page 9, Lines 55-57: Editing issue with the statement "without people without disabilities" This change has been made.

9) Page 11: Table 4 needs some work. It is difficult to read, largely because SES is measured differently across countries, it is treated as a dependent variable instead of a covariate, and the results are presented differently. The table is also missing information that can help the reader to interpret the findings more quickly. For example, denoting significance would be very helpful for the regression results (ORs) with a footnote at the bottom of the table showing how it is denoted?

SES has been removed as an outcome variable, due to inconsistent measurement across countries.

10) Page 10, Table 3: Please consider providing restating the sample size of the individuals (as you did in Table 2) in Table 3. Also, you need to provide the total sample size of the households in Table 3 or somewhere. As far as I can tell, you do not provide the sample size of the households anywhere in the paper except, possibly, in Table 4 on Page 11, Lines 7-9, but these samples do not match the percentages of households with at least one person with a disability you provide in Table 3 on Page 10, Lines 26-29.

The sample size of individuals is now restated in Table 3. The total sample size of households is now presented in Table 2.

11) Page 11; Lines 7-9: As mentioned previously, your household percentages do not match between Tables 3 (Page 10, Lines 26-29) and 4 (Page 11, Lines 7-9). According to the figures in

Table 3, 7.1%, 6.5%, 39.3%, 38.5%, and 10.9% of households in BDG-1, BDG-2, Cameroon, India, and Malawi, respectively, have at least 1 person with a disability. Then, in Table 4, the "household" sample sizes provided (Page 11, Lines 7-9) imply that the percentages of households with disabilities in the sample are about 7%, 12%, 61%, 60%, and 50%, for BDG-1, BDG-2, Cameroon, India, and Malawi, respectively. This could be a simple issue of you not providing enough information about the household sample sizes used in this analysis or it could be an analysis error. It is difficult to tell without more information, but please review and determine how best to let the reader know what is going on here.

The sample size for the household level has been clarified to account for the case-control methodology used in Cameroon, India and Malawi. The sample size for each analysis has been better explained.

12) Page 12, Line 25: Your sample sizes keep changing in these analyses. I assume you are trying to maximize your sample for each analysis, but this changing of sample sizes between analyses makes this very difficult to interpret. I would suggest choosing a single sample size for all analyses or use imputation if it is justifiable.

The sample size for the household level has been clarified to account for the case-control methodology used in Cameroon, India and Malawi. The sample size for each analysis has been better explained. An error in the sample size for Table 5, Bangladesh-2 has been corrected.

13) Page 12, Lines 49-end of page : You report that the data is not shown. From where did this information come? Follow up verbatim questions? None of this is discussed in the measures section either. Again, I recognize word limit challenges, but I think it would be appropriate to specify the measure(s) from which these findings were taken and/or how the percentages were assessed. Since this information is actually the focus of your discussion and conclusion, it seems like the data should, in fact, be shown.

The data on reasons why persons with disabilities do not use the same latrine, or collect water, are now included in Table 5.

14) Table 6, Page 14: You are missing sample information for Malawi. Also, again, your sample sizes are changing and you have no footnote to explain it like in Table 5. Please consider a single list-wise sample or imputation. Also please consider denoting significance by including p-values or notation to make this table faster to interpret for the reader.

The sample size is now given in Table 6 and clarified throughout the document. Statistical significance is denoted in Table 7 (previously Table 6) using an asterisk.

15) Page 4, Lines 40-41: I am not entirely sure you answered your third research question. I assume the analyses leading up to and the results in Tables 5 & 6 are the attempt to answer the third research question (factors associated with access to improved WASH among people with disabilities), but I don't see how the outcome variables reflect the definition of improved water/sanitation (with the exception of "access without contact with feces). Perhaps consider revising research question 3.

Research question 3 has been revised to read "which factors predict access to WASH among people with disabilities"

Reviewer: 4

Reviewer Name: Kristen Heitzinger

Institution and Country: Kentucky Department for Public Health, USA Competing Interests: None declared.

This is an interesting article written on an important topic. However, major revisions are needed in order for this manuscript to be ready for publication.

Specific comments:

Page 1, Line 38: Please correct affiliation 6 from "icdr, b" to "icdr,b"
This change has been made.

Page 2, Line 23: The authors state that there was no difference in "episodes of diarrhoea" when (4-week or 1-week) prevalence of diarrhea (not numbers of episodes) was actually compared.

This has been changed to read "frequency of reported diarrhoeal episodes", as per the methods used.

Page 2, Line 17: Please consider replacing "through" with "using" for clarity.

This change has been made.

Page 3, Line 11: Please do not capitalize "state" or "regional".

This change has been made.

Page 4, Line 26: I suggest softening the language from "confirmed by qualitative research" to "supported by qualitative research" because this is more defensible statement given the paucity of research on this topic.

This change has been made.

Page 4, Line 35: I suggest removing the word "both" because three things (disability, access to WASH, and experience of WASH) are noted here.

This change has been made.

Page 4, Lines 36-37: The phrasing is confusing because the household, not the person, is the unit of analysis for this comparison. I would rephrase to "1) whether households containing people with disabilities have different access to WASH compared to households without disabled members" or something similar.

This has been changed to read "1) whether households including people with disabilities have different access to WASH compared to households without disabled members;"

Page 5: Please clarify the details of the methods. For example, the authors state the intention to assess "access to WASH" and "improved WASH," but do not define these terms in the text. Is "access to WASH" defined as having access to both improved water and sanitation? Moreover, there very little discussion of hygiene, thus the outcomes may be more appropriately described as "access to water and sanitation" or "improved water and sanitation."

This has been clarified in the methods

Page 5, Line 6: I suggest rephrasing from "one in each of Cameroon" to "one each in Cameroon" for clarity.

This section has been rewritten to read: "We analysed data from five cross-sectional, population-based studies conducted at the district or regional level that had collected data on disability and access to, and experience of, WASH. These studies were conducted in: Cameroon, Malawi, India, and Bangladesh two discrete studies)."

Page 5, Lines 13-14: Please consider rephrasing from "One of the studies conducted a census" to "One of the studies collected census data to facilitate random sampling of participants" or something similar. In its present form, the connection between the first phrase of this sentence and the second phrase is not very clear.

This section has been rewritten to read: "One of the studies conducted a census, that is, included all individuals living within the area (Bangladesh-2). The remaining four studies utilized cluster sampling to select a random sample of participants."

Page 5, Lines 17-18: Please consider rephrasing from "people identified in the survey to have disabilities" to "people identified to have disabilities in the survey". The phrase is clearer when the descriptor is closer to the noun it describes.

This change has been made.

Page 5, Line 23: Please specify what boards provided ethical approval for this study so that readers can assess whether the necessary approvals were obtained.

The ethical boards that approved each study are now included in the methods

Page 5, Lines 52-57: Please provide more detail regarding who participated in the surveys. For example, who were the "individual members of the household" who participated in the surveys in Cameroon and India and how were they selected? In line 56, who was the proxy for participants who could not communicate independently? Was it the primary caregiver for that person? This information may be helpful to readers in assessing whether the data collected regarding these individuals is likely to be accurate.

"Individual members of the household" has been replaced with "people with disabilities and matched control subjects". We have clarified that the "proxy" was "usually their primary caregiver or the head of the household".

Page 6, "Other data collected" section: Who were the individual members of the household who were surveyed in Cameroon and India? Did all individual members present at the time of the visit respond or was a household representative (i.e. the oldest member) selected as the respondent? Please provide more detail. Additionally, for consistency, I suggest that the heading for this section should either be formatted in the same way as for the "Disability assessment" and "WASH assessment" sections or it could be placed in the paragraph above the "Disability assessment" section.

The individual members of the household has been replaced with "people with disabilities and matched control subjects". The formatting change has been made.

Table 1 comments:

-The response rates are not included for the Bangladesh-2 and Malawi studies. Please include them or indicate that the data are not available.

Response rates have been added for Bangladesh 2 and Malawi

-In the "sampling strategy" row, "Cross-sectional study", as described for Bangladesh-2 and Malawi, is a study design, not a sampling strategy. Please specify what sampling strategies were used for these surveys.

The sampling strategy used in Malawi has been clarified in Table 1

-Please write out abbreviations in footnotes (i.e. WG, EHG, LSHTM, WEDC) to improve clarity.

WG has been spelled out in full in the table. The abbreviations have been written out in full in the footnotes.

Page 8, Lines 5-8: Please specify if any nonresponse data were available. Given the association between age and disability, if information is available on the ages of the members of households that did not participate in the surveys, this information could be used in the svy analysis to improve the accuracy of the disability prevalence estimates.

Non-response data were not available for all datasets, which has been stated as a limitation on page 20 lines 67 - 68.

Page 8, Line 17: Please provide more detail regarding the coding of SES for the Bangladesh-2 survey to enable replication of methods. For example, what data were used to categorize the SES of the households? Were the households grouped into tertiles?

SES data in Bangladesh-2 was pre-coded by interviewers as “ultra-poor”, “poor”, and “non-poor” based on household land and livelihood category. This has been clarified in the methods with a reference provided.

Page 8, Line 18: Was SES considered as a continuous or categorical variable for the households surveyed in Malawi? By “average household income”, was “average annual household income” intended? Please provide more detail regarding how SES was measured for these households.

Annual household income was collected in Malawi as a continuous numerical variable. This has been clarified in the text.

Page 8, Lines 27-28: This language is unclear. Please consider rephrasing from “a control household selected through matching an individual by age-sex-cluster to the person with a disability” to “control households selected by age-sex-cluster matching to a household member who did not have a disability” or something similar.

This has been rewritten to read “In Cameroon and India, these data were available for all households that included a person with a disability, and for age-sex-cluster matched control households that did not include a person with a disability .”

Page 8, Line 32: I suggest replacing “considered” with “tested” or “evaluated” or a similar term because this is what occurred.

“Considered” has been replaced with “evaluated”.

Page 8, Line 36: I suggest replacing “calculated” with “identified” because the predictors (variables) were not calculated.

This change has been made.

Page 9, Line 10: The statement that the “majority of the population were children or young adults” seems difficult to support because of the broad age categories used in the BGD-2 study, which makes up over 75% of the study population. Please consider modifying this statement in order to better reflect the data.

This has been rephrased to read “a large proportion of the sample were children or young adults”

Page 9, Table 2: Please provide a footnote for each of the BGD-2 estimates specifying the age range of the estimate.

This change has been made.

Page 9, Lines 38-41: It seems this section refers to the types of impairment data in Table 3, but it is unclear whether these lines accurately describe the Table 3 data. The Table 3 footnote b, “Type as proportion of people with disabilities amongst those reported to have disabilities, neither prevalence estimate nor mutually exclusive” is very difficult to understand and I suggest rephrasing this. Does this table describe the proportion of disability types among the total number of disabilities (i.e. the total number of disabilities—not people with disabilities—is the denominator)? If so, please consider replacing this footnote with “Proportion of type of impairment out of total number of disabilities” or something similar. Taking this into consideration in the Page 9 text, the units of analysis are disabilities, not people, and therefore the phrase “accounting for over half of the people identified to

have a disability” should be rephrased to “accounting for over half of the disabilities identified” or something similar.

The footnote has been rephrased to read “Functional difficulty reported amongst people classified as having a disability. These figures are not prevalence estimate and the categories are not mutually exclusive.” It is clarified throughout that we are referring to type of functional difficulty among people classified with disabilities. The Page 9 text has been rephrased to read “The most frequent type of functional limitations reported was physical (walking) in each of the five datasets. The proportions of people with disabilities reporting other functional difficulties (hearing, visual or intellectual) varied across the surveys, but were overall broadly similar in four of the five datasets.”

Page 9, Line 43: Please define “age-related” impairment in the methods. If there is frequent overlap between “age-related” impairments with other types of impairments, the authors could consider excluding this category, which would increase the comparability of the results with the other studies.

A definition for “age-related” has been provided. The BGD-1 data allowed only one response for “type” of disability. Therefore these categories are mutually exclusive and the removal of those within the “age-related” category would bias the data.

Page 10, Table 3 comments:

-The inclusion of “(%)” and “proportion” in the title is redundant. Please simplify.

This change has been made

-Given the title, it is unnecessary to include “%” in the bottom row of the table. Please remove.

This change has been made

-Footnote a: Please specify the units of age (i.e. years).

This change has been made

-Footnote b: Please see comment above about clarifying this.

This footnote has been rephrased to read “Functional difficulty reported amongst people classified as having a disability. These figures are not prevalence estimate and the categories are not mutually exclusive.”

-Footnote **: Please specify what age range was included to generate each estimate.

The age ranges for each estimate are now provided in the footnote

Page 11, Table 4 comments:

-The column title “Effect size (OR or difference)” makes it very unclear what result is being shown. I suggest being consistent in providing only ORs or indicating with a footnote which estimates are differences.

This change has been made.

-Please include an effect size for the “socioeconomic status” row.

In response to the previous reviewer, we have taken the socioeconomic status row out of the table and have included the estimates in the text.

-Having no toilet is classified by the WHO/UNICEF Joint Monitoring Programme (JMP) as unimproved sanitation. Please combine these groups or provide justification for why they should be separated. I suggest combining these groups on the basis that this will also simplify interpretation of the results.

No toilet has now been included in the “unimproved sanitation” category.

-Footnote b is unclear. Is “Return” intended to mean “round-trip travel time”? If so, this should be specified.

“Return” is intended to mean “round-trip travel time”. This has been changed.

-The JMP considers access to an improved water source as being within 30 minutes of total travel time. Please consider combining the 15-30 min. category with the less than 15min. category if, per footnote b, these times represent round-trip travel times.

This change has been made.

-Please replace “water facility” with “water source.”

This change has been made.

-The estimate of the effect size for the association between the water source and presence of a household member with a disability (OR=1.1, 95% CI=1.2-1.9) is implausible because the effect size lies outside of the 95% confidence interval. Please verify and correct.

This was a typographical error and has now been corrected.

-Regarding the row “At least one episode of diarrhoea in the last four weeks,” does this refer to any member of the household? If so, please specify this in footnote c or in the methods section.

This has been clarified in footnote c.

-In footnote a, please provide more detail regarding whether average annual household income was used or some other measure.

SES has been removed as an outcome variable and therefore removed from Table 5

-In footnote c, please provide a more complete explanation of the differences in data collection by site. For example, “A 7-day recall period was used in Malawi.” It is unclear what is intended by “individual not household in Cameroon/India.” Please clarify.

This has been clarified.

Page 12, Lines 5-6: This statement is directly contradicted by the results shown in Table 4, because in Cameroon, households with a disabled member were significantly more likely to have an improved water source. Please modify this statement to accurately reflect the study findings.

A typographical error in Table 5 (previously Table 4) has been corrected.

Page 12, Line 7: Please correct the spelling of “Bangaldesh-2”.

This change has been made.

Page 12, Line 7-8: This statement is contradicted by Table 4 because in BGD-2, a time to water source of 15-30min. was significantly associated with a reduced likelihood of being a household with a disabled member. Please modify this statement and/or provide data that demonstrate this statement is true.

The text has been updated to accurately reflect the data.

Page 12, Lines 12-13: The statement that there were no differences in access to an improved water source seems to repeat the first sentence in this paragraph that was not supported by the data in Table 4. Please consider deleting.

This is an error – now corrected –as the first sentence was meant to read sanitation facility.

Page 12, Lines 14-17: If the Table 4 row “at least one episode of diarrhoea in the last four weeks” is accurate, then what is being measured is household prevalence of diarrhea. Please rephrase “no differences in the frequency of reported diarrhoeal episodes” to “no differences in prevalence of reported household diarrhoea” or something similar to clarify this point.

This change has been made.

Page 12, Table 5: I suggest modifying “Can access drinking water without assistance” to “Can access drinking water in the home without assistance” to improve clarity.

This change has been made.

Page 12, Table 5: Please add a “%” to the BGD-2 estimate for consistency with the formatting of the other cells.

This change has been made.

Page 13, Lines 3-6: This statement (“Almost all people with disabilities reported...”) seems to be contradicted by the data shown in Table 5. Per Table 5, in India, only 58.7% of individuals could access drinking water in the home without assistance. In BGD-1, only 80.8% of individuals could use the same bathing source as other household members. It is inaccurate to represent these prevalences as “almost all people” and therefore I recommend modifying this statement.

This has been rephrased to read “Most people with disabilities...”

Page 13, Line 7: The statement that open defecation “did not affect relative intra-household WASH access” does not appear to be supported by the data shown in Table 5, because intra-household comparisons were not made. Please consider removing this statement or modifying it in order to be better supported by the data.

This statement has been removed.

Page 13, Line 9: Please modify “intra-household access” to “intra-household WASH access” for clarity.

This change has been made.

Page 13, Line 10: Please consider modifying the phrase “the three datasets that collected these data” because a dataset cannot collect data.

This has been rephrased to read “included” rather than “collected”.

Page 13, Lines 15-16: It is unclear on what basis the authors conclude that women with disabilities were “generally more likely...to have the same WASH access as other members of the household.” None of the associations between female gender and use of the same facility as other household members are significant for Cameroon, India or Malawi, and one of the associations is in the opposite direction of the other two. Please remove this sentence or modify it so that it is more clearly supported by the data.

This sentence has been rephrased to read: “There was evidence that women with disabilities were more likely than men with disabilities to use the same sanitation facilities as other members of the household in Malawi, and to be able to collect water for themselves in Cameroon and Malawi.”

Page 14, Table 6: By the heading “same facility as other household members”, please specify whether this refers to a sanitation facility or water source.

This clarification has been made.

Page 16, Lines 11-12: The statement that “no differences in access to improved...water sources were observed between households with and without a person with a disability” may be contradicted by the estimate for Cameroon provided in Table 4. Please verify this estimate and correct this sentence accordingly.

This text has been updated to reflect the data.

Page 16, Lines 14-15: It is unclear what data the authors used to support the conclusion that “households with a person with a disability were more likely to have to spend over thirty minutes fetching water than those without at the Cameroonian and Indian sites.” Only the estimate for the Indian site appears to be statistically significant (does not include 1.0) and if the basis for this conclusion is the observation of a positive association, regardless of statistical significance, then the same conclusion should be drawn for the Malawi site.

Statistically significant ($p < 0.001$) differences between estimates are provided in Table 5 for both Cameroon and India to support this statement.

Page 16, Lines 15-16: The statement that households with people with disabilities live further away from water sources appears not to be supported by the data shown in Table 4, as disability was associated with having a >30min. trip to a water source at only one site. Please provide additional data to support this statement or remove it.

It is now clarified that this was apparent in two sites.

Page 16, Lines 24-28: It is unclear what evidence the authors used to support the statement that “people with more severe limitations generally had less equitable access...” In Table 6, severe disability was significantly associated with a lower likelihood of using the same facility as other household members in only one of the three countries where these data were collected. Including the definition of “WASH access” in the methods section would be helpful to clarify how this conclusion was drawn.

This has been rephrased as “Finally, a multivariate regression analysis highlighted that people with more severe limitations generally experienced greater difficulties in access to WASH than those with moderate limitations”

Page 16, Lines 34-35: It is unclear how the authors arrived at the conclusion that “In India...two thirds of households with a disabled member practiced open defecation...”. This does not follow from the 60.5% prevalence of having “no toilet” among households noted in Table 4. Using the data shown in Table 5 ($(274/(274+508))=65\%$), one would draw the conclusion that 65% of individuals with disabilities living in India practice open defecation, but this refers to individuals, not households, and 65% is more accurately described as “nearly two thirds.” Please modify this sentence to more accurately reflect the data shown in the results section.

This sentence has been rephrased as “. In India, for example, almost two thirds of people with disabilities interviewed practiced open defecation, and over half of people with disabilities were unable to do this without coming into direct contact with faeces, which may put them at substantial risk of diarrhoeal disease and hygiene-related stigma”

Page 16, Line 37: Reference 25 (Murray and Lopez) does not support the authors’ statement that “contact with faeces...may put them at substantial risk of diarrhoeal disease and hygiene-related stigma.” Moreover, there are more up-to-date estimates of the global burden of disease attributable to WASH. Please cite this literature or other literature that supports this statement.

The statement has been rewritten as: “These findings are of concern as it is well established that inadequate sanitation increases the risk of diarrhoeal diseases”. The following reference has been

given, to replace the Murray and Lopez reference: “Prüss-Ustün A, Bartram J, Clasen T, Colford JM Jr, Cumming O, Curtis V, Bonjour S, Dangour AD, De France J, Fewtrell L, Freeman MC, Gordon B, Hunter PR, Johnston RB, Mathers C, Mäusezahl D, Medlicott K, Neira M, Stocks M, Wolf J, Cairncross S. Burden of disease from inadequate water, sanitation and hygiene in low- and middle-income settings: a retrospective analysis of data from 145 countries. *Trop Med Int Health*. 2014 Aug;19(8):894-905.”

Page 16, Line 38: The data do not support the authors’ statement that “there was evidence that people with disabilities have more difficulties...accessing [water] within their homes” because the access of water within the home was not compared between individuals with and without disabilities in Table 5. Please clarify what evidence the authors used to support this conclusion.

This statement has been rephrased as: “Furthermore, there was evidence that many people with disabilities had difficulties collecting drinking water themselves and without assistance, and inadequate drinking water is linked to increased risk of diarrhoeal disease (Pruss-Ustun, 2014).”

Page 16, Line 42: Reference 26 (Manz 2005) discusses the long-term health effects of mild dehydration. It does not support the authors’ statement that difficulty in accessing water in the home is “linked to lower water consumption, dehydration, and the potential for increased morbidity.” Please provide additional support for this statement or consider removing it.

This statement has been rephrased as: “Furthermore, there was evidence that many people with disabilities had difficulties collecting drinking water themselves and without assistance, and inadequate drinking water is linked to increased risk of diarrhoeal disease (Pruss-Ustun, 2014).”

Page 16, Lines 49-51: This sentence is unclear, and there is more up-to-date literature describing the effect of improved WASH on the global burden of disease. Please cite this literature (i.e. the 2010 global burden of disease study) and consider rephrasing this sentence as “Several meta-analyses have quantified the effect of improved WASH on the global burden of disease measured in Disability-Adjusted Life Years (DALYs)” or something similar.

The sentence has been rephrased as suggested by the reviewer, and three new references have been included.

Page 16, Line 54: What is intended by the term “quality of WASH access”? Please provide more detail on the aspects of WASH access intended by this term.

This statement has been rephrased as: “However, no previous quantitative studies have estimated the implications of disability for access to improved WASH, and in particular, consideration of the particular challenges facing people with disabilities compared to other household members in WASH access”

Page 16, Lines 57-58: As per my earlier comment, there does not seem to be data to support the conclusion that women with disabilities have better WASH access. Please provide evidence for this conclusion.

This sentence has been modified to reflect the inconclusive evidence on gender.

Page 17, Line 10: The authors describe the binary measure used in the Bangladesh-1 study as “restrictive,” yet the measures used in the other surveys are also binary. Please justify the description of this measure as “restrictive” or consider removing this word from the text.

The word “restrictive” has been removed.

Page 17, Lines 10-11: The data provided do not support the conclusion that the measure of disability used in the Bangladesh-1 study was the reason that the prevalence was lower than in other studies. Please cite literature supporting this conclusion or modify this sentence to read “which may explain the low prevalence” instead of “explaining the low prevalence.”

The sentence has been rephrased to read “which may explain the low prevalence...”

Page 17, Lines 16-20: This sentence is unclear because the “similar trends” were not specified and no evidence was provided to support the idea that differences in the measurement of disability would affect the assessment of disability severity. Please clarify what is meant by “similar trends” and cite support from the literature to support this statement.

“Similar trends” has been replaced with “similar associations of disability with age and gender and similar patterns of disability type”

Page 17, Line 21: Please do not capitalize “state” and “regional” because they are not proper nouns.

This change has been made.

Page 17, Line 24: Please include examples of what aspects of WASH would be considered “indicators of quality access to WASH” to clarify this concept.

The following examples have been included: e.g. can access sanitation facility without contact with faeces, can access drinking water without assistance

Page 17, Line 36-38: As noted in the methods sections, disability was not consistently defined as a reported measure, but also was defined using clinical criteria in Cameroon and India. Please consider revising this sentence accordingly by removing the word “reported.”

Rephrased as “people with severe....”

Page 17, Line 39: Per my comment above, please provide evidence that female gender was predictive of intra-household WASH access.

The mention of gender has been removed in the conclusion.

Page 17, Lines 39-41: The statement that “the quality of WASH access is likely to have been poorer among people with disabilities” is not well supported by the data provided in this manuscript because individuals with and without disabilities were not directly compared. Please consider rephrasing this to read “the quality of WASH access may have been poorer among people with disabilities”.

The revision has been made as suggested by the reviewer.

VERSION 2 – REVIEW

REVIEWER	Rob Bain UNICEF, USA
REVIEW RETURNED	24-Oct-2017

GENERAL COMMENTS	Many thanks for addressing earlier comments Please note the following minor changes for the discussion: "Specifically, these findings promote the inclusion of intra-household access questions in JMP, to better understand and overcome lower quality access to WASH amongst persons with disabilities within their households." I think you either mean that these should be included in the JMP's recommended questions for household surveys. The JMP does not conduct surveys but provides technical recommendations.
---

REVIEWER	Dr. Tarique Md. Nurul Huda
-----------------	----------------------------

	icddr,b, Bangladesh
REVIEW RETURNED	06-Jan-2018

GENERAL COMMENTS	It was a very important manuscript. There is very limited literature available on access to WASH for people with disability. So I appreciate the Authors attempting to address this very important yet neglected topic. Abstract:  1. Line 53 to 59: Are these Odds Ratios adjusted or crude? 2. In the abstract there is now data presented on the objective three mentioned on last paragraph on page 4. 3. Introduction: I am not sure if such long list of the ethical approval board as in page 5 is required given it takes a lot of space. Is there a way to refer to them in any original articles published using these data sets or may be placed in appendix. Methods:  4. Line 170 on page 6: In the Bangladesh-1 survey disability was defined differently from rest of the data sets. So I was wondering what is the reason for even including this data set? The authors points out in line 112 that use of inconsistent definition of disability is a gap in literature. Results:  5. The results might be more organized if they were organized by the research question/objective. 6. On page 12 in line 20 and 21 more likely was used twice in the same sentence. Discussion:  7. On page 20 line 16, the authors discussed about shared sanitation facility. But it is not clear how the discussion is relevant for that paragraph or how it is linked with any of the study findings. 8. On page 20 lines 22 to 28 the authors presents some data but does not really discusses the implications of these findings or explains what these data may mean. 9. Line 97 on page 22: The authors write, "Consequently, the quality of the WASH access is likely to have been poorer among people with disabilities." I am not sure how was this conclusion derived? Was it based on any data presented?
--

VERSION 2 – AUTHOR RESPONSE

Dear Dr Sucksmith,

Many thanks for your feedback and for the opportunity to revise this manuscript for publication in BMJ Open.

We have revised the manuscript in response to the editorial and reviewer comments. Please see our response to reviewers below.

Yours faithfully,

Islay Mactaggart

Response to reviewers

Editor Comments: Please move the list of the ethics committees that approved your study to a supplementary file, and refer to this in the methods section on page 5.

Thank you for this comment. The ethical boards that reviewed the datasets have now been moved to a supplementary file which is referred to on Page 5 line 147.

Reviewer: 1

Reviewer Name: Rob Bain

Institution and Country: UNICEF, USA

Competing Interests: None declared

Many thanks for addressing earlier comments

We are grateful for the time and thought that was taken into providing comments on our previous draft of the manuscript, which have significantly improved the overall paper.

Please note the following minor changes for the discussion:

"Specifically, these findings promote the inclusion of intra-household access questions in JMP, to better understand and overcome lower quality access to WASH amongst persons with disabilities within their households." I think you either mean that these should be included in the JMP's recommended questions for household surveys. The JMP does not conduct surveys but provides technical recommendations.

Thank you for this recommendation. We have updated the discussion to read "Specifically, these findings promote the inclusion of intra-household access questions in the JMP's recommended questions for household surveys, to better understand and overcome lower quality access to WASH amongst persons with disabilities within their households." – page 18, line 51

Reviewer: 2

Reviewer Name: Dr. Tarique Md. Nurul Huda Institution and Country: icddr,b, Bangladesh Competing Interests: None declared

It was a very important manuscript. There is very limited literature available on access to WASH for people with disability. So I appreciate the Authors attempting to address this very important yet neglected topic.

We thank the reviewer for their positive response to this manuscript.

Abstract:

1.Line 53 to 59: Are these Odds Ratios adjusted or crude?

Unfortunately, because some datasets collected information from the household head, and others from the person with a disability directly, we did not adjust for age or sex, and these odds ratios are crude. However Table 7 provides age and sex adjusted correlates of WASH access amongst persons with disabilities for the datasets that we have the relevant information for.

2.In the abstract there is now data presented on the objective three mentioned on last paragraph on page 4.

Thank you for this comment. Line 62, page 2 (last line of the abstract results section) in fact details results against objective three as follows “These difficulties were most marked for people with more severe impairments.”

3.Introduction: I am not sure if such long list of the ethical approval board as in page 5 is required given it takes a lot of space. Is there a way to refer to them in any original articles published using these data sets or may be placed in appendix.

The list of ethical approval boards has now been moved to a Supplementary Information File that is referred to in the text.

Methods:

4.Line 170 on page 6: In the Bangladesh-1 survey disability was defined differently from rest of the data sets. So I was wondering what is the reason for even including this data set? The authors points out in line 112 that use of inconsistent definition of disability is a gap in literature.

We thank the reviewer for this comment. We are of the view that given the limited data available in this important research area, all datasets should be included with the caveats of doing so clearly stated (as in the discussion). We believe that this encourages further debate and discussion regarding the need for consistent disability definitions, and provides evidence as to why this is important.

Results:

5.The results might be more organized if they were organized by the research question/objective.

We thank the reviewer for this comment. We have structured the results so that Objective 1 (whether households including persons with disabilities have different access to WASH compared to households without disabled members) is reported in Table 5; Objective 2 (whether persons with disabilities have different access to WASH compared to other members of their household) is reported in Table 6 and objective 3 (which factors predict access to WASH among persons with disabilities) is reported in Table 7. Prior tables report sample characteristics and therefore we feel that the results do in fact follow the structure of the research question/objectives.

6.On page 12 in line 20 and 21 more likely was used twice in the same sentence.

Thank you for this edit. The erroneous second “more likely” has now been removed from this sentence.

Discussion:

7.On page 20 line 16, the authors discussed about shared sanitation facility. But it is not clear how the discussion is relevant for that paragraph or how it is linked with any of the study findings.

We apologise but it is unclear which paragraph the reviewer is referring to here. We discuss shared sanitation facilities in paragraph 2 of the discussion. We feel that this is very relevant given that we found that households with a person with a disability were more likely to use shared facilities in Bangladesh-2. We would be grateful for clarification on this comment.

8. On page 20 lines 22 to 28 the authors presents some data but does not really discusses the implications of these findings or explains what these data may mean.

We apologise for this oversight. A follow up sentence “This suggests that the use of shared sanitation facilities does not necessarily equate to negative sanitation experience.” has been added at the end of this paragraph (page 18, line 20 of current draft).

9. Line 97 on page 22: The authors write, “Consequently, the quality of the WASH access is likely to have been poorer among people with disabilities.” I am not sure how was this conclusion derived? Was it based on any data presented?

We conclude that the quality of WASH access is likely to have been poorer among people with disabilities in accordance with the results, showing that intra-household quality of WASH access was poorer amongst persons with disabilities in each sample (Table 6).